# ON THE INTERACTION OF BATCH NOISE, ADAPTIVITY, AND COMPRESSION, UNDER $(L_0, L_1)$-SMOOTHNESS: AN SDE APPROACH

## ABSTRACT

Understanding the dynamics of distributed stochastic optimization requires accounting for several major factors that affect convergence, such as gradient noise, communication compression, and the use of adaptive update rules. While each factor has been studied in isolation, their joint effect under realistic assumptions remains poorly understood. In this work, we develop a unified theoretical framework for Distributed Compressed SGD (DCSGD) and its sign variant Distributed SignSGD (DSignSGD) under the recently introduced $(L_0, L_1)$-smoothness condition. Our analysis leverages stochastic differential equations (SDEs), and we show that while standard first- and second-order SDEs might lead to misleading conclusions, carefully incorporating curvature-dependent terms into the drift helps capture the fine-grained interaction between learning rates, gradient noise, compression, and the geometry of the loss landscape. These tools allow us to inspect the dynamics under general gradient noise assumptions, including heavy-tailed and affine-variance regimes, which extend beyond the classical bounded-variance setting. Our results show that normalizing the updates of DCSGD emerges as a natural condition for stability, with the degree of normalization precisely determined by the gradient noise structure, the landscape's regularity, and the compression rate. In contrast, our model predicts that DSignSGD converges even under heavy-tailed noise with standard learning rate schedules, a finding which we empirically verify. Together, these findings offer both new theoretical insights and practical guidance for designing stable and robust distributed learning algorithms.

## 1 INTRODUCTION

Understanding the dynamics of stochastic optimization algorithms is crucial for the success of large-scale machine learning, particularly in the distributed setting where multiple factors simultaneously affect convergence. Modern distributed training pipelines must cope with three intertwined challenges:

1. **Batch noise.** Stochastic gradient methods rely on mini-batches to reduce computational cost, but this introduces variance in the gradient estimates. In practical scenarios, this noise may not only be non-vanishing but can also exhibit complex, heavy-tailed behavior (Simsekli et al., 2019). Such noise has a profound impact on convergence rates, stability, and generalization, especially in nonconvex landscapes.

2. **Communication compression.** In distributed systems, communicating full-precision gradients is often prohibitively expensive. To alleviate this bottleneck, gradient compression techniques such as sparsification, quantization, and sign-based schemes are commonly used. While these methods reduce communication overhead, they alter the optimization dynamics by introducing bias and additional variance (Alistarh et al., 2017). Understanding the trade-off between efficiency and convergence guarantees under compression remains a central question.

3. **Adaptivity.** Many successful optimizers in deep learning, such as Adam, AdaGrad, or SignSGD, incorporate some form of normalization or adaptivity in their update rules. Adaptivity has been empirically shown to mitigate the detrimental effects of noise and ill-conditioning (Safaryan and Richtarik, 2021), yet a rigorous understanding of *why* adaptivity helps in distributed, noisy, and compressed scenarios is still incomplete. In particular, the interaction between adaptivity and the statistical properties of the gradient noise is far from fully understood.

Despite a substantial body of work on each of these components in isolation, their *joint* interplay remains underexplored, especially under realistic assumptions on the loss landscape. Most existing theoretical results rely on *L-smoothness*, i.e., the assumption that the gradient of the objective function is globally Lipschitz continuous (Bubeck et al., 2015). While this simplifies analysis, it fails to capture the complexities of many practical problems, including those encountered in nonconvex optimization for deep learning. In contrast, the $(L_0, L_1)$-*smoothness* condition allows the norm of the Hessian of the loss to grow at most affinely with its gradient norm, sensibly relaxing the aforementioned regularity condition (Zhang et al., 2020b). Similarly, while most of the literature relies on the assumption that the gradient noise is bounded or has bounded variance, more realistic models, such as affine variance and heavy-tailed noise, are increasingly being adopted in recent work.

While the use of SDEs to study optimization algorithms has seen significant growth in recent years (Li et al., 2017)[1], to the best of our knowledge, no prior work has employed SDEs to analyze stochastic optimizers under the $(L_0, L_1)$-smoothness framework. This represents a significant gap, as $(L_0, L_1)$-smoothness has emerged as a realistic alternative to $L$-smoothness for modeling modern nonconvex landscapes. Following the standard approach, we first considered *first-order* SDE approximations, which are naturally well suited to handle a broad range of noise models, e.g., Gaussian, affine, or even heavy-tailed, and historically led to new conceptual (Su et al., 2014) and practical (Jastrzebski et al., 2018) insights. However, these models yield fundamentally wrong conclusions: they do not prescribe any learning rate constraints and incorrectly suggest that constant-stepsize SGD converges unconditionally. A natural next step is to move to *second-order* models. Yet the *classic second-order SDE from the literature* turns out to be even more problematic: it again fails to enforce learning rate restrictions and, due to its curvature-dependent correction, even predicts *accelerated convergence* at large stepsizes where SGD in fact diverges. Beyond this, neither of these models captures the more subtle dynamics that arise under $(L_0, L_1)$-smoothness, where no universal stepsize can guarantee stability. To overcome these limitations, we derive *new SDE approximations* that correctly recover the *standard learning rate restrictions and stability threshold*, and align closely with the dynamics of their respective optimizers. These new SDEs form the foundation of our analysis.

Building on these models, we develop a comprehensive analysis of DCSGD and DSignSGD under $(L_0, L_1)$-smoothness with flexible gradient noise assumptions encompassing affine variance and heavy-tailed noise. In settings already examined in the literature, such as $(L_0, L_1)$-smoothness with affine variance of the noise,[2] our results are consistent with established findings. In previously unexplored regimes, where $(L_0, L_1)$-smoothness, affine variance, heavy-tailed noise, and gradient compression are brought together under a unified framework, our analysis provides novel insights that advance the understanding of the interaction of these factors.

Beyond these specific contributions, our results conceptually and technically establish the importance of deriving proper SDE models for analyzing optimization in deep learning. We summarize our contributions below and provide Table 1 to facilitate comparison with existing results.

**Contributions.** Building on the above motivations, this work makes the following contributions:

1. From a technical perspective, we formally derive new SDE models that correctly capture the learning rate restrictions and stability thresholds *also* under $(L_0, L_1)$-smoothness, which cannot be done by classic SDE models;

2. Proving convergence bounds for the models of DCSGD and DSignSGD under $(L_0, L_1)$-smoothness and more general batch noise assumptions than those commonly used in the literature;

3. Demonstrating that the degree of normalization required for DCSGD to converge is precisely determined by the interplay between the compression rate, the structure of gradient noise, and the smoothness constants of the loss;

4. Highlighting that an *adaptive* method such as DSignSGD converges even under heavy-tailed noise with standard assumptions on the learning rate scheduler, while DCSGD would diverge.

## 2 RELATED WORK

**SDE Approximations in Optimization.** Continuous-time models in the form of (stochastic) differential equations are a well-established tool to study discrete-time optimizers, e.g. (Helmke and Moore,

---

[1]We provide a comprehensive literature review in Section 2.

[2]This setting is studied for Normalized SGD and AdaGrad (Faw et al., 2023; Wang et al., 2023; Chen et al., 2023), and not for DCSGD or DSignSGD.

Table 1: Comparison of existing convergence results for stochastic methods applied to $(L_0, L_1)$-smooth problems. All results are derived for non-convex problems, and the bounds are given in expectation unless stated otherwise. All works assume bounded noise or bounded variance unless stated otherwise. Abbreviations: "HT" = heavy-tailed noise, "Affine var." = affine variance.

| Reference | Dynamics | Noise | | $(L_0, L_1)$-smooth | Compression |
| | | HT | Affine var. | | |
| --- | --- | --- | --- | --- | --- |
| (Zhang et al., 2020b;a) (Zhao et al., 2021) (Crawshaw et al., 2022) (Koloskova et al., 2023) (Li et al., 2023)[1][2] (Hübler et al., 2024) (Li et al., 2024)[1][3] (Gaash et al., 2025)[1][3] | Discrete | ✗ | ✗ | ✓ | ✗ |
| (Faw et al., 2023)[1][2] (Wang et al., 2023)[1][2] (Chen et al., 2023) | Discrete | ✗ | ✓ | ✓ | ✗ |
| (Khirirat et al., 2024) | Discrete | ✗ | ✗ | ✓ | ✓ |
| (Chezhegov et al., 2025)[1][3] | Discrete | ✓ | ✗ | ✓ | ✗ |
| (Compagnoni et al., 2025a) | Continuous | ✓ | ✗ | ✗ | ✓ |
| **This work** | Continuous | ✓ | ✓ | ✓ | ✓ |

[1] High-probability convergence analysis.
[2] Convergence bounds have inverse-power dependence on the failure probability.
[3] Derived for convex problems.

1994; Kushner and Yin, 2003). Recent works also derived differential equations to model SGD under heavy-tailed batch noise (Simsekli et al., 2019), and (Zhou et al., 2020) derived a Lévy-driven stochastic differential equation to model the non-Gaussianity of the noise. Importantly, it was (Li et al., 2017) that first introduced a *rigorous* theoretical framework to derive SDEs that faithfully model the stochastic behavior intrinsic to optimization algorithms widely employed in machine learning. Since then, such SDE-based formulations have been applied across several domains, including *stochastic optimal control* for tuning stepsizes (Li et al., 2017; 2019) and batch sizes (Zhao et al., 2022). Notably, SDEs have been instrumental in analyzing *convergence bounds* and *stationary distributions* (Compagnoni et al., 2023; 2024; 2025b), *scaling laws* (Jastrzebski et al., 2018; Compagnoni et al., 2025b;a), *implicit regularization* effects (Smith et al., 2021; Compagnoni et al., 2023), and *implicit preconditioning* (Xiao et al., 2025; Marshall et al., 2025). We refer the interested reader to (Orvieto and Lucchi, 2019b;a) for a didactic introduction to this topic, especially for how Itô calculus is used in the derivation of these results. Importantly, we highlight that the results derived from SDEs in the literature pertain to these models only, and not the optimizer they approximate. However, SDEs and optimizers are theoretically guaranteed to stay close up to a certain error, and extensive experiments have shown that SDEs do track their respective optimizers on a variety of DL models and datasets (Compagnoni et al., 2025a;b; Xiao et al., 2025; Marshall et al., 2025).

We contribute to this line by highlighting a key gap: both the classic *first*-order and the *second-order* SDEs from the literature can yield conclusions that contradict the discrete-time dynamics of SGD. While this is somewhat expected from a first-order model, it is surprising that a higher-order one also fails, possibly even more catastrophically. While previous studies did derive second-order SDEs for various optimizers (Li et al., 2017; 2019; Luo et al., 2024), they did not exploit them to obtain theoretical insights and thus overlooked these limitations. By contrast, we derive *new SDE models* whose dynamics agree with the respective algorithms better.

**Interplay of noise, compression, and adaptivity under $(L_0, L_1)$-smoothness.** Previous research has extensively studied the effect of batch noise, compression, and adaptivity on the convergence of optimizers. Batch noise significantly influences stochastic gradient algorithms, affecting their convergence speed and stability (Simsekli et al., 2019; Zhang et al., 2020b; Kunstner et al., 2024; Compagnoni et al., 2025b). Noise characteristics such as heavy-tailed distributions have been shown to

profoundly impact the optimization trajectories, necessitating robust algorithmic strategies (Şimşekli et al., 2019; Gorbunov et al., 2021). Compression methods, including unbiased techniques such as sparsification and quantization (Alistarh et al., 2017; Stich et al., 2018; Mishchenko et al., 2024) and biased approaches such as SignSGD (Bernstein et al., 2018; Balles and Hennig, 2018), are critical for reducing communication overhead in distributed training. These compression techniques come with theoretical guarantees under various smoothness assumptions (Alistarh et al., 2017; Gorbunov et al., 2020; Mishchenko et al., 2024; Compagnoni et al., 2025a), and recent results also develop linear-rate or near-optimal behavior under generalized/$(L_0, L_1)$-smoothness (Vankov et al., 2025; Tyurin, 2024). Adaptive methods such as SignSGD normalize gradient elements to cope effectively with large or heavy-tailed gradient noise, thus demonstrating improved empirical robustness (Safaryan and Richtarik, 2021; Compagnoni et al., 2025b;a; Kornilov et al., 2025).

However, most of the works mentioned above rely on restrictive assumptions such as $L$-smoothness, i.e., the $L$-Lipschitz continuity of the gradient. To relax this condition, Zhang et al. (2020b) introduces and empirically validates the $(L_0, L_1)$-*smoothness* assumption, which allows the norm of the Hessian to be bounded by an affine function of the gradient norm, thereby significantly expanding the class of admissible problems. A growing body of work now analyzes (stochastic) *first*-order methods under $(L_0, L_1)$ or more "generalized-smoothness" assumptions, including Clip-SGD and related clipping schemes (Zhang et al., 2020b;c; Koloskova et al., 2023; Reisizadeh et al., 2025; Gorbunov et al., 2025; Vankov et al., 2025; Gaash et al., 2025; Pethick et al., 2025), Normalized SGD and variants with normalization-based schedules (Zhao et al., 2021; Chen et al., 2023; Hübler et al., 2024; Yang et al., 2024), SignSGD (Crawshaw et al., 2022), AdaGrad (Faw et al., 2023; Wang et al., 2023), Adam (Li et al., 2024; Zhang et al., 2024), and SGD (Li et al., 2023). Beyond these, there are accelerated and proximal/mirror-descent developments under generalized or $(L_0, L_1)$-smoothness (Tyurin, 2025; Yu et al., 2025a; Tovmasyan et al., 2025; Yu et al., 2025b), nonlinearly preconditioned methods (Oikonomidis et al., 2025), results on escaping saddle points (Cao et al., 2025), zero-/first-order complexity under generalized smoothness (Lobanov and Gasnikov, 2025), and decentralized/federated formulations with generalized smoothness and local steps (Demidovich et al., 2024; Jiang et al., 2025). For compressed communication, Khirirat et al. (2024) proposed and analyzed a momentum-based variant of normalized EF21-SGD (Richtárik et al., 2021) under bounded variance. Additional generalized-smoothness analyses further connect normalization, compression, and relaxed smoothness guarantees (Lobanov et al., 2024; Tyurin, 2024; Yang et al., 2024).

In summary, while prior works have leveraged SDEs to model optimization dynamics, they have not addressed the interplay of $(L_0, L_1)$-smoothness, affine variance, heavy-tailed noise, and compression. This gap motivates our analysis.

## 3  PRELIMINARIES

**Distributed Setup.**  Let us consider the problem of minimizing an objective function expressed as an average of $N$ functions: $\min_{x \in \mathbb{R}^d} \left[ f(x) \coloneqq \frac{1}{N} \sum_{i=1}^{N} f_i(x) \right]$, where each $f_i : \mathbb{R}^d \to \mathbb{R}$ is lower bounded and twice continuously differentiable, and represents the loss over the local data of the $i$-th agent. In our stochastic setup, each agent only has access to gradient estimates: let $n_i$ be the number of datapoints accessible to agent $i$; at a given $x \in \mathbb{R}^d$, agent $i$ estimates $\nabla f_i(x)$ using a batch of data $\gamma_i \subseteq \{1, \ldots, n_i\}$, sampled uniformly with replacement and uncorrelated from the previously sampled batches. Given the sampling properties above, this estimate, which we denote by $\nabla f_{i,\gamma_i}(x)$, can be modeled as a perturbation of the global gradient: $\nabla f_{i,\gamma_i}(x) = \nabla f(x) + Z_i(x)$.

**Noise assumptions.**  We assume the sampling process and agent configurations are such that, for all $x \in \mathbb{R}^d$ and each agent pair $(i, j)$ with $i \neq j$, $Z_i(x)$ is independent of $Z_j(x)$. Regarding assumptions on the noise structure, we always assume that at each $x \in \mathbb{R}^d$, $Z_i(x)$ is absolutely continuous and has a coordinate-wise symmetric distribution. For context, we highlight that numerous works in the literature assume a much more restrictive assumption, e.g. that $Z_i(x)$ are Gaussian (Ahn et al., 2012; Chen et al., 2014; Mandt et al., 2016; Stephan et al., 2017; Zhu et al., 2019; Wu et al., 2020; Xie et al., 2021), and Li et al. (2017); Mertikopoulos and Staudigl (2018); Raginsky and Bouvrie (2012); Zhu et al. (2019); Mandt et al. (2016); Ahn et al. (2012); Jastrzebski et al. (2018) even assume the covariance matrix of the batch noise to be constant: we refer the reader to Jastrzebski et al. (2018) for the intuition behind this modeling choices. Finally, if we discuss the setting $Z_i(x) \in L^1(\mathbb{R}^d)$, then we assume $\mathbb{E}[Z_i(x)] = 0$. Lastly, if $Z_i(x) \in L^2(\mathbb{R}^d)$, we denote $\Sigma_i(x) := Cov(Z_i(x))$.

Next, we define our two structural assumptions. The first one strictly concerns the global landscape; the second concerns how global landscape features affect the noise distribution of each agent.

**Definition 3.1.** $f$ is $(L_0, L_1)$-smooth $(L_0, L_1 \geq 0)$ if, $\forall x \in \mathbb{R}^d$, $\left\|\nabla^2 f(x)\right\| \leq L_0 + L_1\|\nabla f(x)\|_2$.

**Definition 3.2** (Extension of the assumptions from Schmidt and Roux (2013); Vaswani et al. (2019)). The gradient noise for agent $i$ has *affine* $(\sigma_{0,i}^2, \sigma_{1,i}^2)$-variance if $\|\Sigma_i(x)\|_\infty \leq \sigma_{0,i}^2 + \sigma_{1,i}^2\|\nabla f(x)\|_2^2$. If $\sigma_{1,i} = 0$, the noise has bounded variance.

Finally, we define which compressors we use to reduce the communication costs of gradients.

**Definition 3.3.** An unbiased compressor is a stochastic map $\mathcal{C}_\xi : \mathbb{R}^d \to \mathbb{R}^d$ such that $(a) \mathbb{E}_\xi\left[\mathcal{C}_\xi(x)\right] = x$ and $(b) \mathbb{E}_\xi\left[\|\mathcal{C}_\xi(x) - x\|_2^2\right] \leq \omega\|x\|_2^2$ for some compression rate $\omega \geq 0$.

**SDE approximations.** The following definition presents the most commonly used notion that formalizes the idea that an SDE can be a "reliable surrogate" to model an optimizer. It is drawn from the field of numerical analysis of SDEs (see Mil'shtein (1986)) and it quantifies the disparity between the discrete and the continuous processes.

**Definition 3.4.** A continuous-time stochastic process $(X_t)_{t\in[0,T]}$ is an $\alpha$-order weak approximation of a discrete stochastic process $(x_k)_{k=0}^{\lfloor T/\eta \rfloor}$ if for every polynomial growth function $g$, there exists a positive constant $C$, independent of $\eta$, such that $\max_{k=0,\ldots,\lfloor T/\eta \rfloor} |\mathbb{E}g(x_k) - \mathbb{E}g(X_{k\eta})| \leq C\eta^\alpha$. We will often refer to 1-order and 2-order weak approximations as *first-* and *second*-order SDEs.

To illustrate the difference between a *first*-order and a *second*-order SDE, we present here those of SGD in the single-node case, originally formally derived in Theorem 1 of (Li et al., 2017). As we denote the covariance batch noise with $\Sigma(x) = \frac{1}{n}\sum_{i=1}^n (\nabla f(x) - \nabla f_i(x))(\nabla f(x) - \nabla f_i(x))^T$, the *first*-order SDE of SGD is

$$dX_t = -\nabla f(X_t)dt + \sqrt{\eta}\sqrt{\Sigma(X_t)}dW_t, \tag{1}$$

while the *second*-order one is

$$dX_t = -\nabla f(X_t)dt - \frac{\eta}{2}\nabla^2 f(X_t)\nabla f(X_t)dt + \sqrt{\eta}\sqrt{\Sigma(X_t)}dW_t, \tag{2}$$

where term in purple color characterizes the higher-order SDE. We will revisit these formulations in Sec. 4.3 and show that they both lead to misleading conclusions, motivating our new approximation:

$$dX_t = -\nabla f(X_t)dt + \frac{\eta}{2}\nabla^2 f(X_t)\nabla f(X_t)dt + \sqrt{\eta}\sqrt{\Sigma(X_t)}dW_t. \tag{3}$$

**Optimizers and SDEs.** We study: 1) DCSGD defined as $x_{k+1} = x_k - \frac{\eta}{N}\sum_{i=1}^N \mathcal{C}_{\xi_i}\left(\nabla f_{i,\gamma_i}(x_k)\right)$, with unbiased compressors $\mathcal{C}_{\xi_i}$ with SDE models in Eq. 97–116; 2) DSignSGD defined as $x_{k+1} = x_k - \frac{\eta}{N}\sum_{i=1}^N \text{sign}(\nabla f_{i,\gamma_i}(x_k))$, with SDE models in Eq. 136–144.

## 4 THEORETICAL RESULTS

Recall that, in the continuous-time setup, the dynamics of the iterates is modeled by a stochastic process $X_t$ solution to an SDE model. In this setting, the learning rate is a scalar factor in the SDE influencing both its drift and diffusion. To decouple adaptivity from scheduling, we *parametrize our learning rate as a product*: $\eta\eta_t$. To ensure convergence, we *always* assume $\eta_t$ satisfying the Robbins and Monro (1951) conditions: For $\phi_t^i = \int_0^t (\eta_s)^i ds$, we require $\phi_t^1 \overset{t\to\infty}{\to} \infty$, $\frac{\phi_t^2}{\phi_t^1} \overset{t\to\infty}{\to} 0$. For example, these conditions are met for $\eta_t = \frac{1}{(1+t)^a}$ for $a \in (0,1)\backslash\{\frac{1}{2}\}$, as $\phi_t^1 \overset{t\to\infty}{\sim} \frac{1}{t^{a-1}} \overset{t\to\infty}{\to} \infty$ and $\frac{\phi_t^2}{\phi_t^1} \overset{t\to\infty}{\sim} \frac{1}{t^a} \overset{t\to\infty}{\to} 0$. The values $a \in \{\frac{1}{2}, 1\}$ are possible, and the expressions are more convoluted.

**Overview** Our insights concern the conditions on the learning rate $\eta\eta_t$ for convergence, where $\eta_t$ is a predetermined scheduler. We aim to determine how factors such as compression, noise structure, and adaptivity influence the level of normalization required to guarantee convergence. First, we show how first- and second-order continuous-time models from the literature lead to misleading conclusions, as they fail to capture the stability thresholds of the learning rate of GD. Then, we justify the derivation of new models that capture this aspect of the dynamics. Finally, we present Thm. 4.2 and Thm. 4.3, which are derived under these new formulations and empirically validated in Fig. 1.

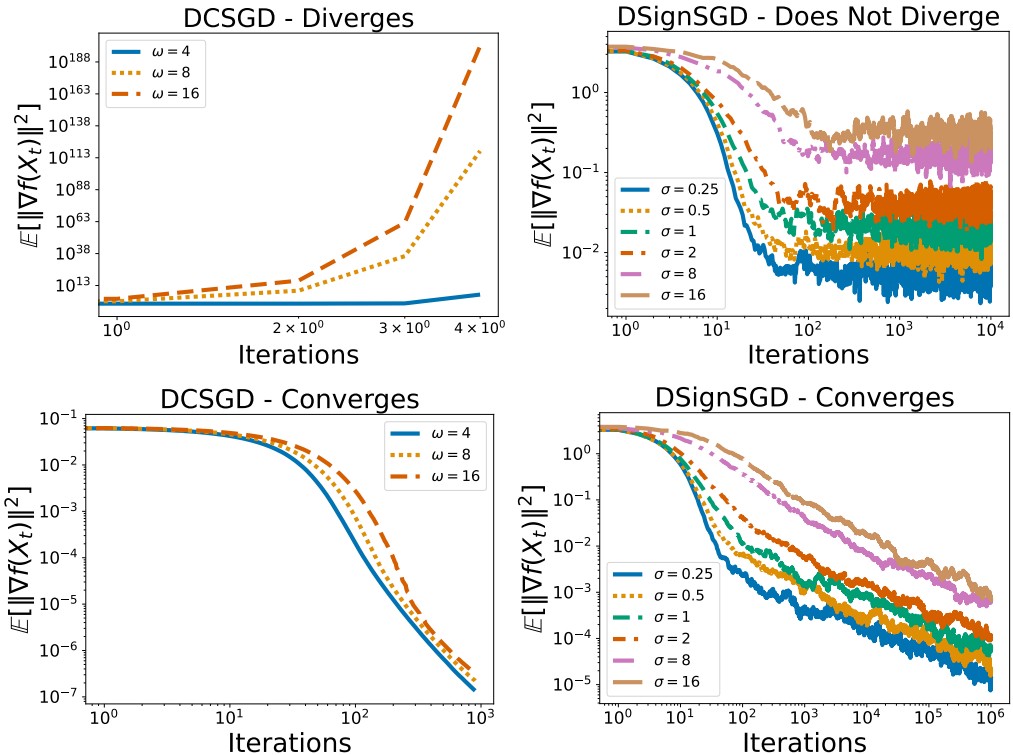

Figure 1: We optimize $f(x) = \frac{\sum_{j=1}^{1000}(x_j)^4}{4}$ with batch noise of variance $\sigma^2 \|\nabla f(x)\|_2^2$ and use *Random Sparsification* for different compression rates $\omega$: as per Thm. 4.2, DCSGD diverges faster and faster for larger values of $\omega$ when the normalization proposed in Eq. 16 **is not employed** (Top-Left) but always converges if it **is employed** (Bottom-Left). We optimize $f(x) = \frac{x^4}{4}$ with batch noise of **unbounded expected value** and for different *scale parameters* $\sigma$: DSignSGD does not converge to $0$ *without* a proper learning rate scheduler as prescribed by Thm. 4.3 (Top-Right), but does converge *with* (Bottom-Right). See Appendix C for all implementation details.

### 4.1 ON THE FAILURE OF CLASSIC FIRST-ORDER SDE MODELS

We start our analysis with a classical approach: As per the literature, we derive a convergence bound for DCSGD from its *first*-order SDE: On the one hand, this result is very insightful and certainly captures important aspects of the dynamics. On the other hand, we quickly figure out its limitations as it fails to capture the fact that Gradient Descent on an $L$-smooth loss only converges if $\eta\eta_t < \frac{2}{L}$.

**Theorem 4.1.** *(DCSGD, unbiased compression, affine variance)* Let $f$ be $(L_0, L_1)$-smooth, and each agent have $(\sigma_{0,i}^2, \sigma_{1,i}^2)$-variance. Define $\overline{\sigma_0^2} := \frac{1}{N}\sum_{i=1}^N \sigma_{0,i}^2$, $\overline{\sigma_1^2} := \frac{1}{N}\sum_{i=1}^N \sigma_{1,i}^2$, $\overline{\sigma_0^2\omega} := \frac{1}{N}\sum_{i=1}^N \sigma_{i,0}^2\omega_i$, and $\overline{\sigma_1^2\omega} := \frac{1}{N}\sum_{i=1}^N \sigma_{i,1}^2\omega_i$. For an arbitrary $\epsilon \in (0,1)$, assume

$$\eta\eta_t < \frac{2\epsilon}{(L_0 + L_1\mathbb{E}[\|\nabla f(X_t)\|_2])\frac{\overline{\omega}+d(\overline{\sigma_1^2\omega}+\overline{\sigma_1^2})}{N} + \frac{L_1 d(\overline{\sigma_0^2}+\overline{\sigma_0^2\omega})}{N}}. \tag{4}$$

*Then, for a random time $\hat{t}$ (independent of $X_t$) with distribution $\frac{\eta_t}{\phi_t^1}$ and $S_0 := f(X_0) - f(X_*)$,*

$$\mathbb{E}\left[\|\nabla f(X_{\hat{t}})\|_2^2\right] \le \frac{1}{(1-\epsilon)\phi_t^1}\left(S_0 + \phi_t^2 \frac{L_0(\overline{\omega}+d(\overline{\sigma_1^2\omega}+\overline{\sigma_1^2})) + L_1 d\left(\overline{\sigma_0^2}+\overline{\sigma_0^2\omega}\right)}{2N}\right) \overset{t\to\infty}{\to} 0. \tag{5}$$

**Intuition:** This result highlights the role of the regularity of the loss landscape and its interaction with both gradient noise and compression. On the one hand, Eq. 4 provides numerous meaningful insights: A larger number of clients $N$ relaxes the restrictions on the learning rate, while an increasing number of trainable parameters $d$, together with larger noise levels $\overline{\sigma}_0$ and $\overline{\sigma}_1$, tighten them; We can notice the intriguing nonlinear interaction between the smoothness constants $L_0$ and $L_1$ with noise constants

$\overline{\sigma_0}$ and $\overline{\sigma_1}$, and even the need for the learning rate to somewhat scale inversely to the expected gradient norm; Gradient compression $\overline{\omega} > 0$ introduces an additional source of adaptivity pressure, independently of the noise structure. Specifically, it shows how the compression rate $\overline{\omega}$ nonlinearly interacts with $L_0, \overline{\sigma_0}, \overline{\sigma_1}$, and $L_1$: In particular, the term $L_1(\overline{\omega} + d(\overline{\sigma_1^2 \omega} + \overline{\sigma_1^2})) > 0$ communicates that stronger compressions of the gradients in a noisy distributed scenario pose constraints on $\eta\eta_t$, especially when $L_1 > 0$. On the other hand, it is important to stress the limitations of the first-order analysis: If we focus on the noiseless and $L$-smooth scenario with no compression, i.e., we set $\sigma_{0,i} = \sigma_{1,i} = \omega_i = L_1 = 0$, we realize that Eq. 4 does not restrict $\eta\eta_t$ in any way, which is clearly unsatisfactory. This highlights the limitations of the first-order SDE approximation in accurately capturing even the most basic learning-rate condition. Finally, we highlight that this bound is not implementable in practice, as constants such as $L_0, L_1, \sigma_{0,i}, \sigma_{1,i}$, and even $\mathbb{E}\left[\|\nabla f(X_t)\|_2\right]$ are not actually known a priori. While this might look like a limitation, this is common in the literature (Gorbunov et al., 2025): These types of results only identify stability regions and show how different factors driving the dynamics influence it.

Similarly, in Theorem B.8, we leverage the first-order SDE of DSignSGD to derive the convergence bound of DSignSGD. While it recovers the results from (Compagnoni et al., 2025a) when $L_1 = \sigma_1 = 0$, it also predicts no restrictions on the learning rate in the noiseless scenario.

### 4.2 Classic Second-Order Models Fail As Well: We Need New Models

In this subsection, we examine how and why both first- and second-order classical models fail to capture this essential aspect of the dynamics. To avoid overloading the discussions with technicalities intrinsic in Itô calculus, we restrict the analysis to the noiseless and single-node case: We use ODEs only to guide the intuition.

**Quadratic Function** In this setting, GD is modeled via ODEs, e.g., $\Sigma = 0$ in Eq. 1 and Eq. 2. We focus on a 1-dimensional quadratic function where the dynamics can be studied tightly and in closed form: For $f(x) = \frac{\lambda x^2}{2}$ for $\lambda > 0$, a discrete GD step is stable only if $\eta < 2/\lambda$. As per Eq. 1, the *first-order ODE* model is

$$dX_t = -\nabla f(X_t)dt = -\lambda X_t dt, \implies f(X_t) = f(X_0)e^{-2\lambda t} \overset{t \to \infty}{\to} 0, \qquad (6)$$

suggesting convergence independently of $\eta$ and missing the stability threshold, which is not realistic. As per Eq. 2, the *second-order ODE from the literature* is

$$dX_t = -\nabla f(X_t)dt - \frac{\eta}{2}\nabla^2 f(X_t)\nabla f(X_t)dt \implies f(X_t) = f(X_0)e^{-2\lambda\left(1+\frac{\lambda\eta}{2}\right)t} \overset{t \to \infty}{\to} 0, \qquad (7)$$

thus missing the threshold *and* predicting *faster convergence* as $\eta$ increases. This is inconsistent with discrete GD, where large $\eta$ causes divergence, making this result even more puzzling.

**Comparison With Discrete-Time Analysis** Here, we take a step back and closely compare the dynamics of the loss function in discrete-time with that in continuous time as prescribed by the ODEs of GD. Consider GD with constant stepsize $\eta > 0$:

$$x_{t+1} = x_t - \eta\nabla f(x_t). \qquad (8)$$

Using a second-order Taylor expansion around $x_t$ along the GD step gives

$$f(x_{t+1}) - f(x_t) = -\eta\|\nabla f(x_t)\|^2 + \frac{\eta^2}{2}\nabla f(x_t)^\top \nabla^2 f(x_t)\,\nabla f(x_t) + O_{x_t}(\eta^3). \qquad (9)$$

However, the first-order ODE of GD implies that

$$df(X_t) = -\|\nabla f(X_t)\|_2^2 dt. \qquad (10)$$

We immediately notice that this ODE describing the dynamics of the loss function in continuous time is completely missing the second-order information highlighted in purple color. The natural step is to shift to the second-order ODE, which implies that

$$df(X_t) = -\|\nabla f(X_t)\|_2^2 dt - \frac{\eta}{2}\nabla f(X_t)^\top \nabla^2 f(X_t)\nabla f(X_t)dt. \qquad (11)$$

While this ODE of the loss *does* incorporate some second-order information highlighted in purple color, we notice that its sign is flipped with respect to that of the discrete dynamics in Eq. 9. This flipped sign is exactly the factor responsible for the failures of this second-order ODE.

Table 2: Comparison of the learning rate product constraints $\eta\eta_t$ derived from classic SDEs (left column) and our SDEs (right column). Each row corresponds to the theorem pairs: DCSGD Thm. 4.1 vs. Thm. 4.2), and DSignSGD (Thm. B.8 vs. Thm. 4.3). Below, $G := (L_0 + L_1 \mathbb{E}[\|\nabla f(X_t)\|_2])$.

| Setting | Classic SDEs | Our SDEs |
|---|---|---|
| DCSGD | $\dfrac{2\epsilon}{G\frac{\overline{\omega}+d(\overline{\sigma_1^2}\omega+\overline{\sigma_1^2})}{N}+\frac{L_1 d\left(\overline{\sigma_0^2}+\overline{\sigma_0^2}\omega\right)}{N}}$ | $\dfrac{2\epsilon}{G\left(\mathbf{1}+\frac{\overline{\omega}+d(\overline{\sigma_1^2}\omega+\overline{\sigma_1^2})}{N}\right)+\frac{L_1 d\left(\overline{\sigma_0^2}+\overline{\sigma_0^2}\omega\right)}{N}}$ |
| DSignSGD | $\frac{\ell_\nu}{K}$ s.t. $K = \frac{L_1 d\sigma_{\mathcal{H},1}}{2N}$ | $\frac{\ell_\nu}{K}$ s.t. $K = \frac{L_1 d\sigma_{\mathcal{H},1}}{2N} + \sqrt{d}\left(L_0 + L_1\right) M_\nu$ |

**Deriving a New Model: An Ansatz Approach.** Therefore, we understand that choosing the right model for the iterates is critical to capture the aspects of the dynamics under analysis. Inspired by a classic approach in mathematical physics, we propose an *ansatz* for an ODE of the iterates of GD and look for one that models the loss dynamics more closely. For a real number $\alpha$, we propose:

$$dX_t = -\nabla f(X_t)dt + \alpha\nabla^2 f(X_t)\nabla f(X_t)dt, \tag{12}$$

which implies that the loss dynamics is driven by

$$df(X_t) = -\|\nabla f(X_t)\|_2^2 dt + \alpha\nabla f(X_t)^\top \nabla^2 f(X_t)\nabla f(X_t)dt. \tag{13}$$

To match the discrete dynamics of the loss in Eq. 9, we need $\alpha = \frac{\eta}{2}$. Therefore, we get that

$$dX_t = -\nabla f(X_t)dt + \frac{\eta}{2}\nabla^2 f(X_t)\nabla f(X_t)dt, \tag{14}$$

is our candidate ODE for GD: We formalize this in Section A, and Theorem A.5 extends this formalization to the stochastic setting. The key observation is that while our new model for the iterates is only a *first*-order one, it induces an ODE for the loss function that is a *second*-order model for the dynamics of the discrete-time loss of GD. Importantly, in the quadratic case studied above, it implies that

$$f(X_t) = f(X_0)e^{-2\lambda\left(1-\frac{\lambda\eta}{2}\right)t}, \tag{15}$$

which, consistently with GD, converges only if $\eta < \frac{2}{\lambda}$. Finally, we refer the interested reader to Section A.3.2 where we compare the modeling properties of all three ODEs on a quartic function: We find that both classic models predict unconditional convergence of GD, while ours does capture the instability of GD if the learning rate does not scale inversely to the norm of the iterates, in accordance to the discrete dynamics of GD.

**Conclusion:** This analysis suggests that a higher order of a continuous-time model does not necessarily translate into it better modeling the discrete-time dynamics, not even in the simplest cases, and even less in the $(L_0, L_1)$-smoothness setting. In particular, we find that appropriate **first**-order SDEs are better than both the classic first- **and** second-order models when it comes to accurately capturing the stability of the optimizers.

### 4.3 RESULTS DERIVED VIA OUR SDES

In this subsection, we report the convergence bounds for newly derived models of DCSGD and DSignSGD. Compared to standard first- and second-order models, our proposed models reveal the interaction between learning rate schedules, loss landscape, batch noise, and compression in a way that is consistent with the discrete dynamics of known cases in the literature. Before presenting these results, Table 2 summarizes how the constraint on $\eta\eta_t$ changes when moving from leveraging the classic SDEs to ours. The orange color indicates terms that *only* appear due to the use of our SDEs.

**Theorem 4.2.** *(DCSGD, unbiased compression, affine variance) Let $f$ be $(L_0, L_1)$-smooth, and each agent have $(\sigma_{0,i}^2, \sigma_{1,i}^2)$-variance. Define $\overline{\sigma_0^2} := \frac{1}{N}\sum_{i=1}^N \sigma_{0,i}^2$, $\overline{\sigma_1^2} := \frac{1}{N}\sum_{i=1}^N \sigma_{1,i}^2$, $\overline{\sigma_0^2}\omega := \frac{1}{N}\sum_{i=1}^N \sigma_{i,0}^2\omega_i$, and $\overline{\sigma_1^2}\omega := \frac{1}{N}\sum_{i=1}^N \sigma_{i,1}^2\omega_i$. For an arbitrary $\epsilon \in (0, 1)$, assume*

$$\eta\eta_t < \frac{2\epsilon}{(L_0 + L_1\mathbb{E}[\|\nabla f(X_t)\|_2])\left(\mathbf{1}+\frac{\overline{\omega}+d(\overline{\sigma_1^2}\omega+\overline{\sigma_1^2})}{N}\right)+\frac{L_1 d\left(\overline{\sigma_0^2}+\overline{\sigma_0^2}\omega\right)}{N}}. \tag{16}$$

*Then, for a random time $\hat{t}$ (independent of $X_t$) with distribution $\frac{\eta_t}{\phi_t^1}$, we have that*

$$\mathbb{E}\left[\|\nabla f(X_{\hat{t}})\|_2^2\right] \leq \frac{1}{(1-\epsilon)\phi_t^1}\left(f(X_0) - f(X_*) + \phi_t^2\frac{\eta(L_0+L_1)d(\overline{\sigma_0^2}+\overline{\sigma_0^2\omega})}{2N}\right) \overset{t\to\infty}{\Rightarrow} 0. \quad (17)$$

**Intuition:** The interpretation of this result is fully in line with that of Theorem 4.1. However, we notice that the baseline term $1$ is crucial as it allows us to recover the standard stepsize schedule bound derived under $L$-smoothness, i.e., $\eta\eta_t < \frac{2}{L_0}$, when we set $\sigma_{0,i} = \sigma_{1,i} = \omega_i = L_1 = 0$. Additionally, it ensures consistency with the noiseless or affine-variance case: $i$) In the noiseless setup $\overline{\sigma_0} = \overline{\sigma_1} = 0$, normalizing the update step naturally emerges as a condition for convergence, in accordance with (Gorbunov et al., 2025); $ii$) When $L_1\overline{\sigma_1^2} > 0$, stronger adaptivity is required, in line with insights derived from analyses of related methods (Wang et al., 2023; Chen et al., 2023). Finally, we highlight that all insights encompassing compression and affine variance under $(L_0, L_1)$-smoothness are **novel**. Although these results do not yield a closed-form rule for choosing an optimal learning rate, they reveal how the right amount of normalization necessary to ensure convergence is dictated jointly by the compression rate, the variance structure of the noise, and the geometry of the landscape. Our result provides a principled stability condition: just as classic analyses of $L$-smooth losses require $\eta < \frac{2}{L}$ to ensure convergence, our bound plays the analogous role for DCSGD in the presence of compression and affine variance. In this sense, this result offers practitioners concrete guidance on when and how to stabilize DCSGD in challenging regimes.

**DSignSGD, structured noise, unbounded expected value.** To provide informative results for the convergence of DSignSGD under heavy-tailed batch noise, we additionally assume structured noise following a student-$t$ distribution: $\nabla f_{\gamma_i}(x) = \nabla f(x) + \sqrt{\Sigma_i}Z_i$ s.t. $Z_i \sim t_\nu(0, I_d)$, $\nu$ are the d.o.f, and *scale matrices*[3] $\Sigma_i = \mathrm{diag}(\sigma_{1,i}^2, \cdots, \sigma_{d,i}^2)$. Note that if $\nu = 1$, the *expected value* of $Z_i$ is *unbounded*, thus modeling much more pathological noise than simple affine $(\sigma_0^2, \sigma_1^2)$-variance. For our analysis, we further define $M_\nu := \sup_{x\in\mathbb{R}}\{\Xi_\nu'(x)\} > 0$ and $\ell_\nu := 2\Xi_\nu'(0) > 0$, where $\Xi_\nu(x) := x\frac{\Gamma\left(\frac{\nu+1}{2}\right)}{\sqrt{\pi\nu}\Gamma\left(\frac{\nu}{2}\right)}{}_2F_1\left(\frac{1}{2}, \frac{\nu+1}{2}; \frac{3}{2}; -\frac{x^2}{\nu}\right)$, and ${}_2F_1(a, b; c; x)$ denotes the hypergeometric function. Importantly, observe that $\frac{1}{2} + \Xi_\nu(x)$ is the CDF of a Student's t-distribution with $\nu$ degrees of freedom.

**Theorem 4.3.** *Let $f$ be $(L_0, L_1)$-smooth, $\Sigma_i \leq \sigma_{max,i}^2$, $\sigma_{\mathcal{H},1}$ be the harmonic mean of $\{\sigma_{max,i}\}$, and $K := \left(\frac{L_1 d\sigma_{\mathcal{H},1}}{2N} + \sqrt{d}(L_0+L_1)M_\nu\right)$. Then, for a scheduler $\eta\eta_t < \frac{\ell_\nu}{K}$, a random time $\tilde{t}$ (independent of $X_t$) with distribution $\frac{\eta_t\ell_\nu - \eta_t^2\eta K}{\phi_t^1\ell_\nu - \phi_t^2\eta K}$, and $S_0 := f(X_0) - f(X_*)$, we have that*

$$\mathbb{E}\left[\|\nabla f(X_{\tilde{t}})\|_2^2\right] \leq \frac{\sigma_{\mathcal{H},1}}{\phi_t^1\ell_\nu - \phi_t^2\eta K}\left(S_0 + \phi_t^2\eta(L_0+L_1)d\left(\frac{1}{2N} + \frac{M_\nu}{\sigma_{\mathcal{H},1}\sqrt{d}}\right)\right) \overset{t\to\infty}{\Rightarrow} 0. \quad (18)$$

**Intuition:** Higher noise levels, captured by $\sigma_{max,i}$, and heavier tails, captured by the degrees of freedom $\nu$, both tighten the upper bound on $\eta\eta_t$. This effect is further amplified by large values of $L_0, L_1$, and by the dimensionality $d$ of the parameter space. In contrast to DCSGD (see Eq. 16), DSignSGD does not require $\eta\eta_t$ to scale inversely with the gradient norm: its adaptive design already incorporates a form of normalization. The crucial difference from the first-order SDE analysis (Theorem B.8) is the appearance of an additional baseline term, $\sqrt{d}(L_0+L_1)M_\nu$. As a consequence, setting $\sigma_{max,i} = 0$ no longer eliminates the restriction on $\eta\eta_t$; rather, it yields the bound $\eta\eta_t < \frac{1}{\sqrt{d}(L_0+L_1)}$. These findings are confirmed in the right column of Figure 1.

# 5 CONCLUSION

In this paper, we provided the first application of SDEs to $(L_0, L_1)$-smooth problems, deriving the first convergence guarantees for the models of DCSGD and DSignSGD under such conditions, coupled with flexible batch noise assumptions. From a *technical* perspective, we exposed a fundamental limitation of the classic *first-* and *second-order* SDEs: although widely used in the literature, they fail to capture essential aspects of the dynamics. In particular, they do not enforce learning rate constraints, predict qualitatively wrong behaviors such as unconditional convergence or even spurious acceleration, and miss the fact that under $(L_0, L_1)$-smoothness, no fixed stepsize is universally stable. To overcome these issues, we introduced *new SDEs* that faithfully track the discrete dynamics of the respective optimizers, recover the standard learning rate restrictions and stability threshold on known settings, and enable novel theoretical and practical insights in unexplored ones.

---

[3]These are *not* covariance matrices, but we use the same notation to facilitate comparability.

**Practical Insights** From a *practical* perspective, our analysis clarifies the role of adaptivity in ensuring convergence of stochastic optimizers. On one hand, an adaptive method such as DSignSGD converges even under heavy-tailed noise with *unbounded* expectation. On the other hand, normalizing the updates for DCSGD emerges naturally as a strategy to ensure convergence, especially when either the compression rate $\overline{\omega}$ or the variance parameter $\overline{\sigma_1^2}$ is positive. Importantly, our analysis — treating compression and affine variance together within $(L_0, L_1)$-smoothness — is novel, and it shows that the *appropriate* normalization level of the gradients is set by the joint influence of compression, noise structure, and landscape geometry, yielding concrete guidance on when and how to stabilize DCSGD in difficult regimes. Taken together, these findings help explain the empirical success of adaptive methods in deep learning: their updates are, to a significant extent, normalized, counteracting the destabilizing effects of ill-conditioned landscapes and large, possibly heavy-tailed noise.

Our contribution is intentionally foundational: rather than proposing new optimizers, we build a rigorous, unified framework that captures the joint effects of noise, compression, and adaptivity for distributed methods under $(L_0, L_1)$-smoothness. We view this work as a basis for future extensions (e.g., heterogeneous clients, error-feedback, and general biased compressors) and for subsequent analyses that further systematize stochastic optimization. More broadly, we believe this is only a first step in harnessing *second*-order information in SDEs for optimization, and anticipate that further developments along this direction will yield even deeper insights into the dynamics of modern stochastic optimizers, such as SGD with Momentum, NAG, AdamW, and RMSprop.

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

# Appendix

## CONTENTS

## A  NEW ODEs AND SDEs FOR GD AND SGD

### A.1  COMPARISON WITH DISCRETE-TIME ANALYSIS

In this section, we closely compare the dynamics of the loss function in discrete-time with that in continuous time as prescribed by the ODEs of GD. Consider GD with constant stepsize $\eta > 0$:

$$x_{t+1} = x_t - \eta \nabla f(x_t). \tag{19}$$

Using a second-order Taylor expansion around $x_t$ gives

$$f(x_{t+1}) - f(x_t) = -\eta \|\nabla f(x_t)\|^2 + \tfrac{\eta^2}{2} \nabla f(x_t)^\top \nabla^2 f(x_t) \, \nabla f(x_t) + O_{x_t}(\eta^3). \tag{20}$$

However, the first-order ODE of GD implies that

$$df(X_t) = -\|\nabla f(X_t)\|_2^2 dt. \tag{21}$$

We immediately notice that this ODE describing the dynamics of the loss function in continuous time is completely missing the second-order information highlighted in purple color. The natural step is to shift to the second-order ODE, which implies that

$$df(X_t) = -\|\nabla f(X_t)\|_2^2 dt - \tfrac{\eta}{2} \nabla f(X_t)^\top \nabla^2 f(X_t) \nabla f(X_t) dt. \tag{22}$$

While this ODE of the loss does incorporate some second-order information highlighted in purple color, we notice that its sign is flipped with respect to that of the discrete dynamics in Eq. 20. This flipped sign is exactly the factor responsible for the failures of this second-order ODE.

**Deriving a New Model: An Ansatz Approach.** Therefore, we understand that choosing the right model is critical to capture the aspects of the dynamics under analysis. Inspired by a classic approach in mathematical physics, we propose an ansatz for an ODE of the iterates of GD and look for one that models the loss dynamics more closely. For a real number $\alpha$, we propose:

$$dX_t = -\nabla f(X_t)dt + \alpha\eta\nabla^2 f(X_t)\nabla f(X_t)dt, \tag{23}$$

which implies that the loss dynamics is driven by

$$df_t = -\|\nabla f(X_t)\|_2^2 dt + \frac{\eta}{2}\nabla f(X_t)^\top \nabla^2 f(X_t)\nabla f(X_t)dt, \tag{24}$$

To match the discrete dynamics of the loss in Eq. 20, we need $\alpha = \frac{1}{2}$. Therefore, we get that

$$dX_t = -\nabla f(X_t)dt + \frac{\eta}{2}\nabla^2 f(X_t)\nabla f(X_t)dt, \tag{25}$$

is our new candidate ODE for GD.

## A.2 NEW MODELS

First, we define two new models for GD and SGD. Then, we introduce a technical lemma and proceed to prove that our new models are **first**-order models for (S)GD.

**Definition A.1.** Based on the discussion above, we define the new ODE model for GD:

$$dX_t = -\nabla f(X_t)dt + \tfrac{\eta}{2}\nabla^2 f(X_t)\nabla f(X_t)dt, \tag{26}$$

and the new SDE model for SGD:

$$dX_t = -\nabla f(X_t)dt + \tfrac{\eta}{2}\nabla^2 f(X_t)\nabla f(X_t)dt + \sqrt{\eta}\sqrt{\Sigma(X_t)}dW_t. \tag{27}$$

*Remark* A.2. Notice that, contrary to the second-order ODE and SDE from the literature, there is a $+$ rather than a $-$ in front of the $\frac{\eta}{2}\nabla^2 f(X_t)\nabla f(X_t)$. The same logic of flipping the sign has to be applied to all classic second-order ODEs and SDEs to obtain the same benefits as those obtained on GD and SGD.

**Theorem A.3.** *Under the dynamics $\dot{x} = F(x)$ such that $F \in C^3(\mathbb{R})$, fix $t$. One has the expansion*

$$x(t+\eta) = x + \eta F + \frac{\eta^2}{2}F'F + \frac{\eta^3}{6}\left(F''F^2 + (F')^2 F\right) + O(\eta^4),$$

*where all derivatives of $F$ are with respect to $x$, evaluated at $x(t)$.*

*Proof.* By Taylor's theorem about $t$,

$$x(t+\eta) = x(t) + \eta x'(t) + \frac{\eta^2}{2}x''(t) + \frac{\eta^3}{6}x'''(t) + O(\eta^4).$$

Note that:

$$x'(t) = F(x(t)), \quad x''(t) = F'(x(t))F(x(t)), \quad x'''(t) = F''(x(t))F(x(t))^2 + \left(F'(x(t))\right)^2 F(x(t)).$$

$\square$

**Theorem A.4** (ODE approximations of Gradient Descent)**.** *Consider gradient descent (GD) with constant stepsize $\eta > 0$. The following ODEs are all weak-approximations of GD:*

 1. *The first-order approximation from the literature:*

$$dX_t = -\nabla f(X_t)\,dt. \tag{28}$$

 2. *The second-order approximation from the literature:*

$$dX_t = -\nabla f(X_t)\,dt - \frac{\eta}{2}\,\nabla^2 f(X_t)\,\nabla f(X_t)\,dt. \tag{29}$$

 3. *Our newly proposed first-order approximation:*

$$dX_t = -\nabla f(X_t)\,dt + \frac{\eta}{2}\,\nabla^2 f(X_t)\,\nabla f(X_t)\,dt. \tag{30}$$

*Proof.* For simplicity, we consider gradient descent in one dimension, as generalizing to higher dimensions follows the same steps:

$$x_{k+1} = x_k - \eta f'(x_k).$$

We now seek a flow of the form

$$F(x) = -f'(x) + \alpha f'(x) f''(x),$$

and just substitute in the expressions in the previous result. Then, we will study the error as a function of $\alpha$. Note that we want to compute

$$x(t+\eta) = x + \eta F + \frac{\eta^2}{2} F'F + \frac{\eta^3}{6}(F''F^2 + (F')^2 F) + O(\eta^4).$$

We have:

$$F = -f' + \alpha f'f'', \qquad F' = -f'' + \alpha((f'')^2 + f'f'''), \qquad F'' = -f''' + \alpha(3f''f''' + f'f'''').$$

So

$$x(t+\eta) = x + \eta\Big(-f' + \alpha f'f''\Big)$$

$$+ \frac{\eta^2}{2}\Big[f'f'' - \alpha(f'^2 f''' + 2f'(f'')^2) + \alpha^2(f'^2 f''f''' + f'(f'')^3)\Big]$$

$$+ \frac{\eta^3}{6}\Big[-(f'^2 f''' + f'(f'')^2)$$

$$+ \alpha(f'^3 f'''' + 7f'^2 f''f''' + 3f'(f'')^3)$$

$$- \alpha^2(2f'^3 f''f'''' + f'^3(f''')^2 + 11f'^2(f'')^2 f''' + 3f'(f'')^4)$$

$$+ \alpha^3(f'^3(f'')^2 f'''' + f'^3 f''(f''')^2 + 5f'^2(f'')^3 f''' + f'(f'')^5)\Big]$$

$$+ O(\eta^4).$$

Assume now that $\alpha = \beta\eta$, we get

$$x(t+\eta) = x - \eta f'$$

$$+ \eta^2\left(\beta + \tfrac{1}{2}\right) f'f''$$

$$- \frac{\eta^3}{6}\Big[(3\beta + 1) f'^2 f''' + (6\beta + 1) f'(f'')^2\Big]$$

$$+ O(\eta^4),$$

For $\alpha = 0$ we get gradient flow and hence

$$x(t+\eta) = x - \eta f' + \tfrac{1}{2}\eta^2 f'f'' - \tfrac{1}{6}\eta^3\left(f'^2 f''' + f'(f'')^2\right) + O(\eta^4),$$

which is the first-order ODE from the literature.

For $\alpha = -\eta/2$

$$x(t+\eta) = x - \eta f' + \eta^3\left(\tfrac{1}{12} f'^2 f''' + \tfrac{1}{3} f'(f'')^2\right) + O(\eta^4),$$

which is the second-order ODE from the literature.

Finally, for $\alpha = \eta/2$,

$$x(t+\eta) = x - \eta f' + \eta^2 f'f'' - \tfrac{5}{12}\eta^3 f'^2 f''' - \tfrac{2}{3}\eta^3 f'(f'')^2 + O(\eta^4),$$

which is our newly proposed first-order ODE.

$$\square$$

The following theorem formalizes that our new SDE model from Eq. 27 is formally a first-order weak approximation for SGD.

**Theorem A.5** (SDE approximations of Stochastic Gradient Descent). *Consider stochastic gradient descent with constant stepsize $\eta > 0$. Its continuous-time approximations are given by the following SDEs:*

1. *The first-order approximation from the literature:*

$$dX_t = -\nabla f(X_t)\, dt + \sqrt{\eta \Sigma(X_t)}dW_t. \tag{31}$$

2. *The second-order approximation from the literature:*

$$dX_t = -\nabla f(X_t)\, dt - \frac{\eta}{2}\, \nabla^2 f(X_t)\, \nabla f(X_t)\, dt + \sqrt{\eta \Sigma(X_t)}dW_t. \tag{32}$$

3. *Our newly proposed first-order approximation:*

$$dX_t = -\nabla f(X_t)\, dt + \frac{\eta}{2}\, \nabla^2 f(X_t)\, \nabla f(X_t)\, dt + \sqrt{\eta \Sigma(X_t)}dW_t. \tag{33}$$

Here is the formal proof:

*Proof.* We work under the framework and assumptions of Section D. Let $(x_k)_{k \geq 0}$ denote the SGD iterates with constant stepsize $\eta > 0$:

$$x_{k+1} = x_k - \eta\, g(x_k, \xi_{k+1}),$$

where $\{\xi_k\}_{k \geq 1}$ are i.i.d. random variables and $g(\cdot, \xi)$ is an unbiased stochastic gradient estimator:

$$\mathbb{E}[g(x, \xi)] = \nabla f(x), \qquad \mathrm{Cov}(g(x, \xi)) = \Sigma(x).$$

We write the gradient noise as

$$\zeta(x, \xi) := g(x, \xi) - \nabla f(x),$$

so that $\mathbb{E}[\zeta(x, \xi)] = 0$ and $\mathbb{E}[\zeta(x, \xi)\zeta(x, \xi)^\top] = \Sigma(x)$, with bounded moments up to order 3 (consistent with (Li et al., 2017) and Assumption D).

Throughout, we fix $0 < \eta < 1$ and condition on $x_k = x$.

**Step 1: One-step moments of SGD.** Define the one-step increment of SGD by

$$\bar{\Delta} := x_{k+1} - x_k = -\eta\, \nabla f(x_k) - \eta\, \zeta(x_k, \xi_{k+1}).$$

Conditioned on $x_k = x$, we thus have

$$\bar{\Delta} = -\eta\, \nabla f(x) - \eta\, \zeta,$$

where we write $\zeta := \zeta(x, \xi_{k+1})$.

Writing components as $\bar{\Delta}_i$, $f_i = \partial_i f$, and $\Sigma_{ij}$ for the $(i, j)$-th entry of $\Sigma(x)$, we get:

$$\mathbb{E}[\bar{\Delta}_i \mid x_k = x] = -\eta\, \mathbb{E}\big[f_i(x) + \zeta_i \mid x_k = x\big] = -\eta\, f_i(x),$$

$$\mathbb{E}[\bar{\Delta}_i \bar{\Delta}_j \mid x_k = x] = \eta^2\, \mathbb{E}\big[(f_i(x) + \zeta_i)(f_j(x) + \zeta_j) \mid x_k = x\big]$$
$$= \eta^2\big(f_i(x)f_j(x) + \Sigma_{ij}(x)\big).$$

Moreover, for any $s \geq 3$ and indices $i_1, \dots, i_s$, we have

$$\mathbb{E}\Big[\prod_{\ell=1}^{s} \bar{\Delta}_{i_\ell}\, \Big|\, x_k = x\Big] = O(\eta^s) = O(\eta^3),$$

thanks to the bounded higher moments of $\zeta$ and Assumption D. In particular, there exists $K_4 \in G$ such that

$$\mathbb{E}\Big[\prod_{\ell=1}^{s} \big|\bar{\Delta}_{i_\ell}\big|\, \Big|\, x_k = x\Big] \leq K_4(x)\, \eta^3 \leq K_4(x)\, \eta^2,$$

for all $0 < \eta < 1$ and all $s \geq 3$. Hence condition (4) in Theorem D.4 holds.

**Step 2: One-step moments of the SDE family.** We now put the three candidate SDE models in Theorem A.5 into the template of Lemma D.3 and Theorem D.4.

Fix $\alpha \in \mathbb{R}$ and consider the SDE

$$dX_t = b_\alpha(X_t)\,dt + \sqrt{\eta}\,\sigma(X_t)\,dW_t, \tag{34}$$

with

$$b_\alpha(x) := -\nabla f(x) + \alpha\,\eta\,\nabla^2 f(x)\,\nabla f(x), \qquad \sigma(x)\sigma(x)^\top = \Sigma(x).$$

We will later set

$$\alpha = 0,\ -\tfrac{1}{2},\ \tfrac{1}{2}$$

to recover the three SDEs in the statement.

Let $X_0 = x$ and define the one-step increment

$$\Delta := X_\eta - x,$$

with components $\Delta_i$. By Lemma D.3 applied to equation 34, we have

$$\mathbb{E}[\Delta_i] = b_{\alpha,i}(x)\,\eta + \frac{1}{2}\sum_{j=1}^{d} b_{\alpha,j}(x)\,\partial_j b_{\alpha,i}(x)\,\eta^2 + O(\eta^3), \tag{35}$$

$$\mathbb{E}[\Delta_i\Delta_j] = \Big[b_{\alpha,i}(x)b_{\alpha,j}(x) + \big(\sigma\sigma^\top\big)_{ij}(x)\Big]\eta^2 + O(\eta^3)$$

$$= \Big[b_{\alpha,i}(x)b_{\alpha,j}(x) + \Sigma_{ij}(x)\Big]\eta^2 + O(\eta^3), \tag{36}$$

$$\mathbb{E}\Big[\prod_{\ell=1}^{s}\Delta_{i_\ell}\Big] = O(\eta^3), \qquad \forall s \geq 3. \tag{37}$$

All functions above belong to $G$ by Assumption D.

Now plug $b_\alpha(x) = -\nabla f(x) + \alpha\eta\,\nabla^2 f(x)\,\nabla f(x)$ into these expressions. Write

$$h(x) := \nabla^2 f(x)\,\nabla f(x), \qquad h_i(x) = \sum_{j=1}^{d}\partial_{ij}f(x)\,\partial_j f(x).$$

Then

$$b_{\alpha,i}(x) = -f_i(x) + \alpha\,\eta\,h_i(x).$$

For the *first* moment, using equation 35, we expand to order $\eta^2$:

$$\mathbb{E}[\Delta_i] = \Big(-f_i(x) + \alpha\eta\,h_i(x)\Big)\eta + \frac{1}{2}\sum_{j=1}^{d}\Big(-f_j(x) + O(\eta)\Big)\partial_j\Big(-f_i(x) + O(\eta)\Big)\eta^2 + O(\eta^3)$$

$$= -\eta f_i(x) + \alpha\,\eta^2 h_i(x) + \frac{1}{2}\sum_{j=1}^{d}f_j(x)\,\partial_j f_i(x)\,\eta^2 + O(\eta^3)$$

$$= -\eta f_i(x) + \Big(\alpha + \tfrac{1}{2}\Big)h_i(x)\,\eta^2 + O(\eta^3).$$

In particular,

$$\mathbb{E}[\Delta_i] - \mathbb{E}[\bar{\Delta}_i] = \Big(\alpha + \tfrac{1}{2}\Big)h_i(x)\,\eta^2 + O(\eta^3) = O(\eta^2). \tag{38}$$

For the *second* moment, from equation 36 we note that

$$b_{\alpha,i}(x)b_{\alpha,j}(x) = \big(-f_i(x) + O(\eta)\big)\big(-f_j(x) + O(\eta)\big) = f_i(x)f_j(x) + O(\eta),$$

hence

$$\mathbb{E}[\Delta_i\Delta_j] = \big(f_i(x)f_j(x) + \Sigma_{ij}(x)\big)\eta^2 + O(\eta^3), \tag{39}$$

so that

$$\mathbb{E}[\Delta_i\Delta_j] - \mathbb{E}[\bar{\Delta}_i\bar{\Delta}_j] = O(\eta^3) \leq K_2(x)\,\eta^2,$$

for some $K_2 \in G$ and all $0 < \eta < 1$. Similarly, combining equation 37 with the bound on higher moments of $\bar{\Delta}$ from Step 1, we obtain for $s \geq 3$

$$\left| \mathbb{E} \prod_{\ell=1}^{s} \Delta_{i_\ell} - \mathbb{E} \prod_{\ell=1}^{s} \bar{\Delta}_{i_\ell} \right| = O(\eta^3) \leq K_3(x)\, \eta^2$$

for some $K_3 \in G$, and we have already seen that

$$\mathbb{E} \prod_{\ell=1}^{s} |\bar{\Delta}_{i_\ell}| \leq K_4(x)\, \eta^2$$

for appropriate $K_4 \in G$. Thus, all four conditions of Theorem D.4 are satisfied for equation 34, for any fixed $\alpha \in \mathbb{R}$.

**Step 3: Weak order and identification of the three SDEs.** By Theorem D.4, for any fixed $\alpha \in \mathbb{R}$ the SDE

$$dX_t = b_\alpha(X_t)\, dt + \sqrt{\eta}\, \sigma(X_t)\, dW_t$$

is an *order 1 weak approximation* of the SGD recursion, in the sense of Definition D.2.

It remains to identify the three specific choices of $\alpha$ that correspond to the SDEs in the statement.

- **Case $\alpha = 0$.** Then $b_0(x) = -\nabla f(x)$, so the SDE equation 34 becomes

  $$dX_t = -\nabla f(X_t)\, dt + \sqrt{\eta\, \Sigma(X_t)}\, dW_t,$$

  which is exactly the *first*-order SDE approximation from the literature in Eq. equation 31. By Theorem D.4, this is an order 1 weak approximation of SGD.

- **Case $\alpha = -\frac{1}{2}$.** Then

  $$b_{-1/2}(x) = -\nabla f(x) - \frac{\eta}{2}\, \nabla^2 f(x)\, \nabla f(x),$$

  and equation 34 becomes

  $$dX_t = -\nabla f(X_t)\, dt - \frac{\eta}{2}\, \nabla^2 f(X_t)\, \nabla f(X_t)\, dt + \sqrt{\eta\, \Sigma(X_t)}\, dW_t,$$

  which is exactly Eq. equation 32, the classical *second*-order SDE from the literature. In this case, equation 38 shows that

  $$\mathbb{E}[\Delta_i] - \mathbb{E}[\bar{\Delta}_i] = O(\eta^3),$$

  so the drift matches to one order higher; combined with the analysis in (Li et al., 2017), this yields a second-order weak approximation. In particular, it is also an order 1 weak approximation.

- **Case $\alpha = \frac{1}{2}$.** Then

  $$b_{1/2}(x) = -\nabla f(x) + \frac{\eta}{2}\, \nabla^2 f(x)\, \nabla f(x),$$

  and equation 34 becomes

  $$dX_t = -\nabla f(X_t)\, dt + \frac{\eta}{2}\, \nabla^2 f(X_t)\, \nabla f(X_t)\, dt + \sqrt{\eta\, \Sigma(X_t)}\, dW_t,$$

  which is exactly our new SDE model in Eq. equation 33. As shown above, all four conditions of Theorem D.4 hold, so this SDE is also an *order 1 weak approximation* of SGD.

This proves that all three SDEs listed in Theorem A.5 are weak approximations of SGD in the sense of Definition D.2: the first and third are first-order weak approximations, while the second one is the classical second-order stochastic modified equation from the literature.

$\square$

### A.3 COMPARING ODEs - AN INSIGHT PERSPECTIVE

In this section, we showcase how models from the literature fail to properly model the dynamics of GD, especially regarding the constraints on the learning rate to ensure convergence. In contrast, we show that our model is in accordance with GD.

#### A.3.1 QUADRATIC FUNCTION

For didactic reasons, we now compare the proofs for a convergence bound on the loss value $f(x)$ when the loss is a 1-dimensional convex quadratic function $\frac{\lambda x^2}{2}$. **To avoid overloading the proof with technicalities intrinsic in Itô calculus**, we restrict the analysis to the noiseless and single-node case. The *first*-order ODE is

$$dX_t = -\nabla f(X_t)dt = -\lambda X_t dt, \tag{40}$$

which implies that

$$df(X_t) = -2\lambda f(X_t)dt \implies f(X_t) = f(X_0)e^{-2\lambda t} \overset{t\to\infty}{\to} 0, \tag{41}$$

somewhat implying that GD converges independently of the constant $L$ and of the learning rate $\eta$. Much differently, the *second*-order ODE *from the literature* is

$$dX_t = -\nabla f(X_t)dt - \frac{\eta}{2}\nabla^2 f(X_t)\nabla f(X_t)dt, \tag{42}$$

which implies that

$$df(X_t) = -\|\nabla f(X_t)\|_2^2 dt - \frac{\eta}{2}\nabla f(X_t)^\top \nabla^2 f(X_t)\nabla f(X_t)dt = -2\lambda f(X_t)dt - \frac{\eta}{2}\lambda X_t^\top \lambda\lambda X_t \tag{43}$$

$$= -2\lambda\left(1 + \frac{\lambda\eta}{2}\right)f(X_t)dt \implies f(X_t) = f(X_0)e^{-2\lambda\left(1+\frac{\lambda\eta}{2}\right)t} \overset{t\to\infty}{\to} 0, \tag{44}$$

which is also inconsistent with the discrete-time analysis since we get convergence for any $\eta > 0$.

Now, we try to leverage our new ODE derived in Theorem A.4 and get that:

$$df(X_t) = -\|\nabla f(X_t)\|_2^2 dt + \frac{\eta}{2}\nabla f(X_t)^\top \nabla^2 f(X_t)\nabla f(X_t)dt = -2\lambda f(X_t)dt + \frac{\eta}{2}\lambda X_t^\top \lambda\lambda X_t \tag{45}$$

$$= -2\lambda\left(1 - \frac{\lambda\eta}{2}\right)f(X_t)dt \implies f(X_t) = f(X_0)e^{-2\lambda\left(1-\frac{\lambda\eta}{2}\right)t} \overset{t\to\infty}{\to} 0, \tag{46}$$

which only converges if $\eta < \frac{2}{\lambda}$. This is consistent with the analysis in discrete time.

**Conclusion:** First of all, it is immediately apparent that while *first*-order approximations may lead to relevant insights, they prevent us from having a full picture. Second, we demonstrated that the classic *second*-order SDE also led us to results that are inconsistent with the discrete-time analysis. Finally, our model provides a qualitatively faithful description of the true GD dynamics.

#### A.3.2 QUARTIC FUNCTION

Here, we compare the three ODEs listed above as they describe the optimization of a quartic function $f(x) = \frac{x^4}{4}$: We find that the classic ones both fail. First of all, a single step of gradient descent with stepsize $\eta$ reads

$$x_{k+1} = x_k - \eta\nabla f(x_k) = x_k - \eta x_k^3,$$

meaning that if $\eta > \frac{2}{x_k^2}$ the dynamics *explodes*. In particular,

$$\frac{f(x_{k+1}) - f(x_k)}{\eta} = -x_k^6 + \frac{3}{2}\eta x_k^8 + O(\eta^2). \tag{47}$$

Using the first-order ODE, we get that

$$dX_t = -X_t^3 dt \implies f(X_t) = \frac{1}{4(2t + X_0^{-2})^2} \tag{48}$$

This model predicts universal convergence with a polynomial rate, but it does *not* capture the exploding behaviour observed in GD. Using the second-order ODE, we get that

$$dX_t = -X_t^3 dt - \frac{3\eta}{2}X_t^5 dt \implies df(X_t) = -X_t^6 dt - \frac{3\eta}{2}X_t^8 dt, \tag{49}$$

from which we understand that since the additional term is *negative*, this ODE suggests *faster convergence* for larger $\eta$. Using our new ODE, we get that

$$dX_t = -X_t^3 dt + \tfrac{3\eta}{2} X_t^5 dt \implies df(X_t) = -X_t^6 dt + \tfrac{3\eta}{2} X_t^8 dt, \tag{50}$$

which matches the dynamics of the loss of GD up to order 2. Importantly, it captures the phenomenon that the learning rate $\eta$ needs to scale inversely to the norm of the iterates for GD to converge.

**Conclusion.** On the quartic loss, the first-order ODE predicts convergence for all $\eta$, missing the instability. The second-order ODE from the literature predicts *accelerated convergence* for larger $\eta$, in direct contradiction with GD. In contrast, our new ODE reproduces the key phenomenon: the learning rate $\eta$ needs to scale inversely to the norm of the iterates for GD to converge. Hence, our model provides a qualitatively faithful description of the true GD dynamics.

### A.4 DIFFUSION APPROXIMATION FOR THE LOSS IN SGD

In this section, we propose an alternative approach to the derivation of a continuous-time model for SGD. Rather than modeling the iterates and use the Itô Lemma to study the SDE of the loss function, we try a new approach: We directly investigate the possibility of directly modeling the dynamics of the loss. Consider stochastic gradient descent (SGD) with constant stepsize $\eta > 0$:

$$x_{t+1} = x_t - \eta g_t, \qquad g_t = \nabla f(x_t) + \zeta_t, \tag{51}$$

where $f : \mathbb{R}^d \to \mathbb{R}$ is smooth, $\zeta_t$ is the gradient noise satisfying

$$\mathbb{E}[\zeta_t \mid x_t] = 0, \qquad \mathrm{Cov}(\zeta_t \mid x_t) = \Sigma(x_t).$$

We study the dynamics of the *loss process* $Y_t := f(x_t)$.

**Step 1. Taylor expansion of the loss.** Using a second-order Taylor expansion around $x_t$, for $h = -\eta g_t$ we have

$$\begin{aligned} f(x_{t+1}) &= f(x_t + h) \\ &= f(x_t) + \nabla f(x_t)^\top h + \tfrac{1}{2} h^\top \nabla^2 f(x_t) h + O(\|h\|^3). \end{aligned} \tag{52}$$

Substituting $h = -\eta g_t$ gives

$$f(x_{t+1}) - f(x_t) = -\eta \nabla f(x_t)^\top g_t + \tfrac{\eta^2}{2} g_t^\top \nabla^2 f(x_t) g_t + O(\eta^3). \tag{53}$$

**Step 2. Expansion of stochastic terms.** Expanding with $g_t = \nabla f(x_t) + \zeta_t$ yields

$$\begin{aligned} f(x_{t+1}) - f(x_t) = -\eta \|\nabla f(x_t)\|^2 &- \eta \nabla f(x_t)^\top \zeta_t \\ &+ \tfrac{\eta^2}{2} \nabla f(x_t)^\top \nabla^2 f(x_t) \nabla f(x_t) \tag{54} \\ &+ \tfrac{\eta^2}{2} \zeta_t^\top \nabla^2 f(x_t) \zeta_t + \eta^2 \nabla f(x_t)^\top \nabla^2 f(x_t) \zeta_t + O(\eta^3). \tag{55} \end{aligned}$$

**Step 3. Drift and volatility.** Taking the conditional expectation given $x_t$,

$$\begin{aligned} \mathbb{E}[f(x_{t+1}) - f(x_t) \mid x_t] = -\eta \|\nabla f(x_t)\|^2 & \\ + \tfrac{\eta^2}{2} \nabla f(x_t)^\top \nabla^2 f(x_t) \nabla f(x_t) &+ \tfrac{\eta^2}{2} \mathrm{tr}\left(\nabla^2 f(x_t) \Sigma(x_t)\right) + O(\eta^3). \end{aligned} \tag{56}$$

The stochastic fluctuations arise from the linear terms in $\zeta_t$,

$$-\eta \nabla f(x_t)^\top \zeta_t + \eta^2 \nabla f(x_t)^\top \nabla^2 f(x_t) \zeta_t,$$

whose leading-order contribution is

$$-\eta \nabla f(x_t)^\top \zeta_t.$$

This term has conditional variance

$$\mathrm{Var}\left(-\eta \nabla f(x_t)^\top \zeta_t \mid x_t\right) = \eta^2 \nabla f(x_t)^\top \Sigma(x_t) \nabla f(x_t).$$

**Step 4. Continuous-time limit.** Rescaling time by $s = t\eta$ and letting $\eta \to 0$, the increments equation 55 converge in distribution to the diffusion

$$dY_s = \left(-\|\nabla f(X_s)\|^2 + \tfrac{\eta}{2}\nabla f(X_s)^\top \nabla^2 f(X_s)\nabla f(X_s) + \tfrac{\eta}{2}\operatorname{tr}\left(\nabla^2 f(X_s)\Sigma(X_s)\right)\right)ds + G(X_s)\,dW_s,$$
(57)

where $W_s$ is a standard Brownian motion and the scalar volatility $G(x)$ is defined by

$$G(x)^2 = \nabla f(x)^\top \Sigma(x)\,\nabla f(x).$$
(58)

Interestingly, this SDE is the same one that one gets by applying Itô's Lemma on $f(X_t)$ under the dynamics of our newly proposed SDE in Eq. 27, which consolidates the intuition that our model properly captures the dynamics of SGD faithfully.

## B  THEORETICAL RESULTS

**Assumptions and notation.** In line with (Compagnoni et al., 2025a), we assume that the stochastic gradient of the $i$-th agent is given by $\nabla f_{\gamma_i}(x) = \nabla f(x) + Z_i(x)$, where $Z_i(x)$ denotes the gradient noise and $Z_i(x)$ is independent of $Z_j(x)$ for $i \neq j$. If $Z_i(x) \in L^1(\mathbb{R}^d)$, we assume $\mathbb{E}[Z_i(x)] = 0$, and if $Z_i(x) \in L^2(\mathbb{R}^d)$, we assume $Cov(Z_i(x)) = \Sigma_i(x)$ (we omit the size of the batch $\gamma$ unless relevant) s.t. $\sqrt{\Sigma_i(x)}$ is bounded, Lipschitz, satisfies affine growth, and together with its derivatives, it grows at most polynomially fast (Definition 2.5 in Malladi et al. (2022)). Importantly, we assume that all $Z_i(x)$ have a smooth and bounded probability density function whose derivatives are all integrable: A common assumption in the literature is for $Z_i(x)$ to be Gaussian Ahn et al. (2012); Chen et al. (2014); Mandt et al. (2016); Stephan et al. (2017); Zhu et al. (2019); Wu et al. (2020); Xie et al. (2021): See Jastrzebski et al. (2018) for the justification why this could be the case. Differently, our assumption allows for heavy-tailed distributions such as the Student's t. It is important to point out that Li et al. (2017); Mertikopoulos and Staudigl (2018); Raginsky and Bouvrie (2012); Zhu et al. (2019); Mandt et al. (2016); Ahn et al. (2012); Jastrzebski et al. (2018) use a Gaussian noise with a constant covariance matrix to model batch noise.

### B.1  DISTRIBUTED SGD

#### B.1.1  FIRST ORDER SDE

The following is the *first*-order SDE model of DSGD (see Theorem 3.2 in Compagnoni et al. (2025a)). Let us consider the stochastic process $X_t \in \mathbb{R}^d$ defined as the solution of

$$dX_t = -\nabla f(X_t)dt + \sqrt{\frac{\eta}{N}}\sqrt{\hat{\Sigma}(X_t)}dW_t,$$
(59)

where $\hat{\Sigma}(x) := \frac{1}{N}\sum_{i=1}^N \Sigma_i(x)$ is the average of the covariance matrices of the $N$ agents.

**Theorem B.1.** *Let $f$ be $(L_0, L_1)$-smooth, $\|\Sigma_i(x)\|_\infty < \sigma_{0,i}^2 + \sigma_{1,i}^2\|\nabla f(x)\|_2^2$, the learning rate scheduler $\eta_t$ s.t. $\phi_t^i = \int_0^t (\eta_s)^i ds$, $\phi_t^1 \overset{t\to\infty}{\to} \infty$, $\frac{\phi_t^2}{\phi_t^1} \overset{t\to\infty}{\to} 0$, $\overline{\sigma_0^2} := \frac{1}{N}\sum_{i=1}^N \sigma_{0,i}^2$, and $\overline{\sigma_1^2} := \frac{1}{N}\sum_{i=1}^N \sigma_{1,i}^2$. Then, for $0 < \epsilon < 1$,*

$$\eta\eta_t < \frac{2N\epsilon}{d\left(\overline{\sigma_1^2}L_0 + \overline{\sigma_0^2}L_1 + L_1\overline{\sigma_1^2}\mathbb{E}\left[\|\nabla f(X_t)\|_2\right]\right)},$$
(60)

*and for a random time $\hat{t}$ with distribution $\frac{\eta_t}{\phi_t^1}$, we have that*

$$\mathbb{E}\left[\|\nabla f(X_{\hat{t}})\|_2^2\right] \leq \frac{1}{\phi_t^1(1-\epsilon)}\left(f(X_0) - f(X_*) + \phi_t^2\frac{\eta d(L_0 + L_1)(\overline{\sigma_0^2} + \overline{\sigma_1^2})}{2N}\right) \overset{t\to\infty}{\to} 0.$$
(61)

*Proof.* Using Itô's Lemma and using a learning rate scheduler $\eta_t$ during the derivation of the SDE, we have

$$d(f(X_t) - f(X_*)) = -\eta_t\|\nabla f(X_t)\|_2^2 dt + \mathcal{O}(\text{Noise}) + (\eta_t)^2\frac{\eta}{2N}\text{Tr}(\nabla^2 f(X_t)\tilde{\Sigma}(X_t))dt \quad (62)$$

$$\leq -\eta_t\|\nabla f(X_t)\|_2^2 dt + \mathcal{O}(\text{Noise}) \quad (63)$$

$$+ (\eta_t)^2\frac{\eta(\overline{\sigma_0^2} + \overline{\sigma_1^2}\|\nabla f(X_t)\|_2^2)d(L_0 + L_1\|\nabla f(X_t)\|)}{2N}dt, \quad (64)$$

where we used that $\text{Tr}\left(\nabla^2 f(x)\tilde{\Sigma}(x)\right) \leq d\|\nabla^2 f(x)\|_\infty\|\tilde{\Sigma}(x)\|_\infty$ together with the smoothness and noise assumptions. Importantly, $\mathcal{O}(\text{Noise}) = \sqrt{\tilde{\Sigma}(X_t)}\nabla f(X_t)dW_t$.

**Phase 1:** If $\|\nabla f(X_t)\| \leq 1$, we have that

$$d(f(X_t) - f(X_*)) \leq -\eta_t\|\nabla f(X_t)\|_2^2 dt + (\eta_t)^2\frac{\eta(\overline{\sigma_0^2} + \overline{\sigma_1^2})d(L_0 + L_1)}{2N}dt + \mathcal{O}(\text{Noise}), \quad (65)$$

**Phase 2:** If $\|\nabla f(X_t)\| > 1$, we have

$$d(f(X_t) - f(X_*)) = -\eta_t\|\nabla f(X_t)\|_2^2 dt + \mathcal{O}(\text{Noise}) + (\eta_t)^2\frac{\eta}{2N}\text{Tr}(\nabla^2 f(X_t)\tilde{\Sigma}(X_t))dt \quad (66)$$

$$\leq -\eta_t\|\nabla f(X_t)\|_2^2 dt + \mathcal{O}(\text{Noise}) \quad (67)$$

$$+ (\eta_t)^2\frac{\eta(\overline{\sigma_0^2} + \overline{\sigma_1^2}\|\nabla f(X_t)\|_2^2)d(L_0 + L_1\|\nabla f(X_t)\|)}{2N}dt \quad (68)$$

$$= -\eta_t\|\nabla f(X_t)\|_2^2\left(1 - \frac{\eta_t\eta d}{2N}\left(\overline{\sigma_1^2}L_0 + \overline{\sigma_0^2}L_1 + L_1\overline{\sigma_1^2}\|\nabla f(X_t)\|_2\right)\right)dt \quad (69)$$

$$+ (\eta_t)^2\frac{\eta\overline{\sigma_0^2}dL_0}{2N}dt + \mathcal{O}(\text{Noise}). \quad (70)$$

By taking a worst-case scenario approach, we merge these two bounds into a single one:

$$d(f(X_t) - f(X_*)) \leq -\eta_t\|\nabla f(X_t)\|_2^2\left(1 - \frac{\eta_t\eta d}{2N}\left(\overline{\sigma_1^2}L_0 + \overline{\sigma_0^2}L_1 + L_1\overline{\sigma_1^2}\|\nabla f(X_t)\|_2\right)\right)dt \quad (71)$$

$$+ (\eta_t)^2\frac{\eta d(L_0 + L_1)(\overline{\sigma_0^2} + \overline{\sigma_1^2})}{2N}dt + \mathcal{O}(\text{Noise}). \quad (72)$$

Therefore, for $0 < \epsilon < 1$ we have that if

$$1 - \frac{\eta_t\eta d}{2N}\left(\overline{\sigma_1^2}L_0 + \overline{\sigma_0^2}L_1 + L_1\overline{\sigma_1^2}\|\nabla f(X_t)\|_2\right) < 1 - \epsilon, \quad (73)$$

or, equivalently

$$\eta\eta_t < \frac{2N\epsilon}{d\left(\overline{\sigma_1^2}L_0 + \overline{\sigma_0^2}L_1 + L_1\overline{\sigma_1^2}\|\nabla f(X_t)\|_2\right)}, \quad (74)$$

we have that

$$d(f(X_t) - f(X_*)) \leq -\eta_t\|\nabla f(X_t)\|_2^2(1 - \epsilon)dt + (\eta_t)^2\frac{\eta d(L_0 + L_1)(\overline{\sigma_0^2} + \overline{\sigma_1^2})}{2N}dt + \mathcal{O}(\text{Noise}). \quad (75)$$

Therefore,

$$\eta_t\|\nabla f(X_t)\|_2^2(1 - \epsilon)dt \leq -d(f(X_t) - f(X_*)) + (\eta_t)^2\frac{\eta d(L_0 + L_1)(\overline{\sigma_0^2} + \overline{\sigma_1^2})}{2N}dt + \mathcal{O}(\text{Noise}). \quad (76)$$

Dividing by $1 - \epsilon$, integrating over time, and using the martingality of the noise term under the expected value,

$$\int_0^t \eta_s \mathbb{E}\|\nabla f(X_s)\|_2^2 ds \leq \frac{1}{1-\epsilon} \left( f(X_0) - f(X_*) + \phi_t^2 \frac{\eta d(L_0 + L_1)(\overline{\sigma_0^2} + \overline{\sigma_1^2})}{2N} \right). \tag{77}$$

Dividing by $\phi_t^1$ and by the Law of the Unconscious Statistician, we have that

$$\mathbb{E}\left[\|\nabla f(X_{\hat{t}})\|_2^2\right] \leq \frac{1}{\phi_t^1(1-\epsilon)} \left( f(X_0) - f(X_*) + \phi_t^2 \frac{\eta d(L_0 + L_1)(\overline{\sigma_0^2} + \overline{\sigma_1^2})}{2N} \right) \overset{t \to \infty}{\rightarrow} 0, \tag{78}$$

where $\hat{t}$, is a random time with distribution $\frac{\eta_{\hat{t}}}{\phi_t^1}$.

Finally, for practical reasons, we leverage the distributed setting to tighten the requirements on the learning rate scheduler to make it experimentally viable (see Section C.4 for the details), and require

$$\eta \eta_t < \frac{2N\epsilon}{d\left( \overline{\sigma_1^2} L_0 + \overline{\sigma_0^2} L_1 + L_1 \overline{\sigma_1^2} \mathbb{E}\left[\|\nabla f(X_t)\|_2\right] \right)}. \tag{79}$$

$\square$

### B.1.2 Our New First-Order SDE for DSGD

The following is the *first*-order SDE model of DSGD and is a straightforward generalization of Theorem 3.2 in Compagnoni et al. (2025a) and Remark A.2. Let us consider the stochastic process $X_t \in \mathbb{R}^d$ defined as the solution of

$$dX_t = -\nabla f(X_t)dt + \frac{\eta}{2}\nabla^2 f(X_t)\nabla f(X_t)dt + \sqrt{\frac{\eta}{N}}\sqrt{\hat{\Sigma}(X_t)}dW_t, \tag{80}$$

where $\hat{\Sigma}(x) := \frac{1}{N}\sum_{i=1}^N \Sigma_i(x)$ is the average of the covariance matrices of the $N$ agents.

**Theorem B.2.** *Let $f$ be $(L_0, L_1)$-smooth, $\|\Sigma_i(x)\|_\infty < \sigma_{0,i}^2 + \sigma_{1,i}^2\|\nabla f(x)\|_2^2$, the learning rate scheduler $\eta_t$ s.t. $\phi_t^i = \int_0^t (\eta_s)^i ds$, $\phi_t^1 \overset{t \to \infty}{\rightarrow} \infty$, $\frac{\phi_t^2}{\phi_t^1} \overset{t \to \infty}{\rightarrow} 0$, $\overline{\sigma_0^2} := \frac{1}{N}\sum_{i=1}^N \sigma_{0,i}^2$, and $\overline{\sigma_1^2} := \frac{1}{N}\sum_{i=1}^N \sigma_{1,i}^2$. Then, for $0 < \epsilon < 1$,*

$$\eta \eta_t < \frac{2\epsilon}{L_0 + L_1 \mathbb{E}\left[\|\nabla f(X_t)\|\right] + \frac{d}{N}\left( \overline{\sigma_1^2} L_0 + \overline{\sigma_0^2} L_1 + L_1 \overline{\sigma_1^2} \mathbb{E}\left[\|\nabla f(X_t)\|\right] \right)}, \tag{81}$$

*and for a random time $\hat{t}$ with distribution $\frac{\eta_t}{\phi_t^1}$, we have that*

$$\mathbb{E}\left[\|\nabla f(X_{\hat{t}})\|_2^2\right] \leq \frac{1}{\phi_t^1(1-\epsilon)} \left( f(X_0) - f(X_*) + \frac{\eta \phi_t^2}{2N}(L_0 + L_1)d\overline{\sigma_0^2} \right) \overset{t \to \infty}{\rightarrow} 0. \tag{82}$$

*Proof.* Using Itô's Lemma and using a learning rate scheduler $\eta_t$ during the derivation of the SDE, we have that for $\mathcal{O}(\text{Noise}) = \sqrt{\tilde{\Sigma}(X_t)}\nabla f(X_t)dW_t$,

$$d(f(X_t) - f(X_*)) = -\eta_t\|\nabla f(X_t)\|_2^2 dt + \frac{\eta \eta_t^2}{2}\left(\nabla f(X_t)\right)^\top \nabla^2 f(X_t)\nabla f(X_t)dt \tag{83}$$

$$+ \mathcal{O}(\text{Noise}) + (\eta_t)^2 \frac{\eta}{2N}\text{Tr}(\nabla^2 f(X_t)\tilde{\Sigma}(X_t))dt \tag{84}$$

$$\leq -\eta_t\|\nabla f(X_t)\|_2^2 dt + \frac{\eta \eta_t^2}{2}(L_0 + L_1\|\nabla f(X_t)\|)\|\nabla f(X_t)\|^2 dt \tag{85}$$

$$+ \mathcal{O}(\text{Noise}) + (\eta_t)^2 \frac{\eta(\overline{\sigma_0^2} + \overline{\sigma_1^2}\|\nabla f(X_t)\|_2^2)d(L_0 + L_1\|\nabla f(X_t)\|)}{2N}dt. \tag{86}$$

**Phase 1:** If $\|\nabla f(X_t)\| \leq 1$,

$$d(f(X_t) - f(X_*)) \leq \|\nabla f(X_t)\|_2^2 \left( \eta_t - \frac{\eta \eta_t^2}{2}(L_0 + L_1 \|\nabla f(X_t)\|_2) \left( 1 + \frac{d\overline{\sigma_1^2}}{N} \right) \right) dt \quad (87)$$

$$+ \frac{\eta \eta_t^2}{2N} \cdot (L_0 + L_1) d\overline{\sigma_0^2} dt + \mathcal{O}(\text{Noise}). \quad (88)$$

**Phase** 2: If $\|\nabla f(X_t)\| > 1$, we have

$$d(f(X_t) - f(X_*)) = -\eta_t \|\nabla f(X_t)\|_2^2 dt + \mathcal{O}(\text{Noise}) + (\eta_t)^2 \frac{\eta}{2N} \text{Tr}(\nabla^2 f(X_t) \tilde{\Sigma}(X_t)) dt \quad (89)$$

$$\leq -\eta_t \|\nabla f(X_t)\|_2^2 dt + \frac{\eta \eta_t^2}{2}(L_0 + L_1 \|\nabla f(X_t)\|) \|\nabla f(X_t)\|^2 dt \quad (90)$$

$$+ \mathcal{O}(\text{Noise}) + (\eta_t)^2 \frac{\eta(\overline{\sigma_0^2} + \overline{\sigma_1^2} \|\nabla f(X_t)\|_2^2) d(L_0 + L_1 \|\nabla f(X_t)\|)}{2N} dt \quad (91)$$

$$= -\eta_t \|\nabla f(X_t)\|_2^2 \left[ 1 - \frac{\eta_t \eta}{2} \left[ (L_0 + L_1 \|\nabla f(X_t)\|) \left[ 1 + \frac{d\overline{\sigma_1^2}}{N} \right] + \frac{d\overline{\sigma_0^2} L_1}{N} \right] \right] dt$$

$$+ (\eta_t)^2 \frac{\eta \overline{\sigma_0^2} d L_0}{2N} dt + \mathcal{O}(\text{Noise}). \quad (92)$$

By taking a worst-case scenario approach, we merge these two bounds into a single one:

$$d(f(X_t) - f(X_*)) \leq -\eta_t \|\nabla f(X_t)\|_2^2 \left[ 1 - \frac{\eta_t \eta}{2} \left[ (L_0 + L_1 \|\nabla f(X_t)\|) \left[ 1 + \frac{d\overline{\sigma_1^2}}{N} \right] + \frac{d\overline{\sigma_0^2} L_1}{N} \right] \right] dt$$

$$+ (\eta_t)^2 \frac{\eta}{2N}(L_0 + L_1) d\overline{\sigma_0^2} dt + \mathcal{O}(\text{Noise}). \quad (93)$$

With arguments that follow the same steps we detailed in the proof of Theorem B.1, for $0 < \epsilon < 1$, we have that if

$$\eta \eta_t < \frac{2\epsilon}{L_0 + L_1 \|\nabla f(X_t)\| + \frac{d}{N} \left( \overline{\sigma_1^2} L_0 + \overline{\sigma_0^2} L_1 + L_1 \overline{\sigma_1^2} \|\nabla f(X_t)\|_2 \right)}, \quad (94)$$

by integrating over time and by the Law of the Unconscious Statistician, we have that

$$\mathbb{E}\left[ \|\nabla f(X_{\hat{t}})\|_2^2 \right] \leq \frac{1}{\phi_t^1 (1 - \epsilon)} \left( f(X_0) - f(X_*) + \frac{\eta \phi_t^2}{2N}(L_0 + L_1) d\overline{\sigma_0^2} \right) \overset{t \to \infty}{\to} 0, \quad (95)$$

where $\hat{t}$, is a random time with distribution $\frac{\eta_{\hat{t}}}{\phi_t^1}$.

Finally, for practical reasons, we leverage the distributed setting to tighten the requirements on the learning rate scheduler to make it experimentally viable, and rather require

$$\eta \eta_t < \frac{2\epsilon}{L_0 + L_1 \mathbb{E}\left[ \|\nabla f(X_t)\| \right] + \frac{d}{N} \left( \overline{\sigma_1^2} L_0 + \overline{\sigma_0^2} L_1 + L_1 \overline{\sigma_1^2} \mathbb{E}\left[ \|\nabla f(X_t)\| \right] \right)}. \quad (96)$$

$\square$

## B.2 DISTRIBUTED COMPRESSED SGD WITH UNBIASED COMPRESSION

### B.2.1 FIRST ORDER SDE

The following is the *first*-order SDE model of DCSGD (see Theorem 3.6 in Compagnoni et al. (2025a)). Let us consider the stochastic process $X_t \in \mathbb{R}^d$ defined as the solution of

$$dX_t = -\nabla f(X_t) dt + \sqrt{\frac{\eta}{N}} \sqrt{\tilde{\Sigma}(X_t)} dW_t, \quad (97)$$

where for $\Phi_{\xi_i, \gamma_i}(x) := \mathcal{C}_{\xi_i}\left(\nabla f_{\gamma_i}(x)\right) - \nabla f_{\gamma_i}(x)$

$$\tilde{\Sigma}(x) = \frac{1}{N} \sum_{i=1}^{N} \left( \mathbb{E}_{\xi_i \gamma_i} \left[ \Phi_{\xi_i, \gamma_i}(x) \Phi_{\xi_i, \gamma_i}(x)^\top \right] + \Sigma_i(x) \right). \tag{98}$$

**Theorem B.3.** *Let $f$ be $(L_0, L_1)$-smooth, the learning rate scheduler $\eta_t$ such that $\phi_t^i = \int_0^t (\eta_s)^i ds$, $\phi_t^1 \overset{t\to\infty}{\to} \infty$, $\frac{\phi_t^2}{\phi_t^1} \overset{t\to\infty}{\to} 0$, and $\overline{\sigma^2\omega} := \frac{1}{N}\sum_{i=1}^{N} \sigma_i^2 \omega_i$. Then, for $0 < \epsilon < 1$,*

$$\eta\eta_t < \frac{2N\epsilon}{\overline{\omega}L_0 + \left(\overline{\sigma^2}d + d\overline{\sigma^2\omega}\right)L_1 + \overline{\omega}L_1 \mathbb{E}\left[\|\nabla f(X_t)\|_2\right]}, \tag{99}$$

*and for a random time $\hat{t}$ with distribution $\frac{\eta_t}{\phi_t^1}$, we have that*

$$\mathbb{E}\left[\|\nabla f(X_{\hat{t}})\|_2^2\right] \leq \frac{1}{\phi_t^1(1-\epsilon)} \left( f(X_0) - f(X_*) + \phi_t^2 \frac{\eta(L_0 + L_1)d\left(\overline{\sigma^2} + \overline{\sigma^2\omega}\right)}{2N} \right) \overset{t\to\infty}{\to} 0. \tag{100}$$

*Proof.* Since it holds that

$$\mathbb{E}_{\xi_i, \gamma_i} \|(\mathcal{C}_{\xi_i}\left(\nabla f_{\gamma_i}(x)\right) - \nabla f(x))\|_2^2 \leq \omega_i \|\nabla f(x)\|_2^2 + d\sigma_i^2(\omega_i + 1),$$

we have that for $\mathcal{O}(\text{Noise}) = \sqrt{\tilde{\Sigma}(X_t)}\nabla f(X_t)dW_t$,

$$d(f(X_t) - f(X_*)) = -\eta_t \|\nabla f(X_t)\|_2^2 dt + \mathcal{O}(\text{Noise}) \tag{101}$$

$$+ (\eta_t)^2 \frac{\eta(L_0 + L_1\|\nabla f(X_t)\|_2)}{2N} \left( \frac{1}{N} \sum_{i=1}^{N} \mathbb{E}_{\xi_i, \gamma_i} \|(\mathcal{C}_{\xi_i}\left(\nabla f_{\gamma_i}(x)\right) - \nabla f(x))\|_2^2 \right) dt \tag{102}$$

$$\leq -\eta_t \|\nabla f(X_t)\|_2^2 dt + \mathcal{O}(\text{Noise}) \tag{103}$$

$$+ (\eta_t)^2 \frac{\eta(L_0 + L_1\|\nabla f(X_t)\|_2)}{2N} \left( \overline{\omega}\|\nabla f(X_t)\|_2^2 + \overline{\sigma^2}d + d\overline{\sigma^2\omega} \right) dt. \tag{104}$$

**Phase 1:** If $\|\nabla f(X_t)\|_2 \leq 1$, then we have that

$$d(f(X_t) - f(X_*)) \leq -\|\nabla f(X_t)\|_2^2 \left( \eta_t - \frac{\eta(L_0 + L_1)\overline{\omega}}{2N}(\eta_t)^2 \right) dt \tag{105}$$

$$+ (\eta_t)^2 \frac{\eta(L_0 + L_1)d}{2N} \left( \overline{\sigma^2} + \overline{\sigma^2\omega} \right) dt + \mathcal{O}(\text{Noise}). \tag{106}$$

**Phase 2:** If $\|\nabla f(X_t)\|_2 > 1$, we have that

$$d(f(X_t) - f(X_*)) \leq -\eta_t \|\nabla f(X_t)\|_2^2 dt + \mathcal{O}(\text{Noise}) \tag{107}$$

$$+ (\eta_t)^2 \frac{\eta(L_0 + L_1\|\nabla f(X_t)\|_2)}{2N} \left( \overline{\omega}\|\nabla f(X_t)\|_2^2 + \overline{\sigma^2}d + d\overline{\sigma^2\omega} \right) dt \tag{108}$$

$$\leq -\eta_t \|\nabla f(X_t)\|_2^2 \left( 1 - \frac{\eta_t \eta}{2N} \left( \overline{\omega}L_0 + d\left(\overline{\sigma^2} + \overline{\sigma^2\omega}\right)L_1 + \overline{\omega}L_1\|\nabla f(X_t)\|_2 \right) \right) dt \tag{109}$$

$$+ \eta_t^2 \frac{\eta L_0 d}{2N} \left( \overline{\sigma^2} + \overline{\sigma^2\omega} \right) dt + \mathcal{O}(\text{Noise}). \tag{110}$$

By taking a worst-case scenario approach, we merge these two bounds into a single one. With arguments that follow the same steps we detailed in the proof of Theorem B.1, we have that for $0 < \epsilon < 1$, we have that if

$$\eta\eta_t < \frac{2N\epsilon}{\overline{\omega}L_0 + d\left(\overline{\sigma^2} + \overline{\sigma^2\omega}\right)L_1 + \overline{\omega}L_1\|\nabla f(X_t)\|_2}, \tag{111}$$

by integrating over time and by the Law of the Unconscious Statistician, we have that

$$\mathbb{E}\left[\|\nabla f(X_{\hat{t}})\|_2^2\right] \leq \frac{1}{\phi_t^1(1-\epsilon)} \left( f(X_0) - f(X_*) + \phi_t^2 \frac{\eta(L_0 + L_1)d\left(\overline{\sigma^2} + \overline{\sigma^2\omega}\right)}{2N} \right) \overset{t\to\infty}{\to} 0, \tag{112}$$

where $\hat{t}$, is a random time with distribution $\frac{\eta_{\hat{t}}}{\phi_t^1}$.

Finally, for practical reasons, we leverage the distributed setting to tighten the requirements on the learning rate scheduler to make it experimentally viable, and rather require

$$\eta\eta_t < \frac{2N\epsilon}{\overline{\omega}L_0 + \left(\overline{\sigma^2}d + d\overline{\sigma^2\omega}\right)L_1 + \overline{\omega}L_1\mathbb{E}\left[\|\nabla f(X_t)\|_2\right]}. \tag{113}$$

$\square$

Finally, one can generalize this result to cover the $(\sigma_0^2, \sigma_1^2)$-Variance.

**Theorem B.4.** *Let $f$ be $(L_0, L_1)$-smooth, $\max(\Sigma_i(x)) < \sigma_{i,0}^2 + \sigma_{i,1}^2\|\nabla f(x)\|_2^2$, the learning rate scheduler $\eta_t$ such that $\phi_t^i = \int_0^t (\eta_s)^i ds$, $\phi_t^1 \overset{t\to\infty}{\to} \infty$, $\frac{\phi_t^2}{\phi_t^1} \overset{t\to\infty}{\to} 0$, $\overline{\sigma_0^2} := \frac{1}{N}\sum_{i=1}^N \sigma_{0,i}^2$, $\overline{\sigma_1^2} := \frac{1}{N}\sum_{i=1}^N \sigma_{1,i}^2$, $\overline{\sigma_0^2\omega} := \frac{1}{N}\sum_{i=1}^N \sigma_{i,0}^2\omega_i$, and $\overline{\sigma_1^2\omega} := \frac{1}{N}\sum_{i=1}^N \sigma_{i,1}^2\omega_i$. Then, for $0 < \epsilon < 1$,*

$$\eta\eta_t < \frac{2N\epsilon}{L_0(\overline{\omega} + d(\overline{\sigma_1^2\omega} + \overline{\sigma_1^2})) + L_1 d\left(\overline{\sigma_0^2} + \overline{\sigma_0^2\omega}\right) + L_1(\overline{\omega} + d(\overline{\sigma_1^2\omega} + \overline{\sigma_1^2}))\mathbb{E}\left[\|\nabla f(X_t)\|_2\right]}, \tag{114}$$

*and for a random time $\hat{t}$ with distribution $\frac{\eta_t}{\phi_t^1}$, we have that*

$$\mathbb{E}\left[\|\nabla f(X_{\hat{t}})\|_2^2\right] \leq \frac{1}{(1-\epsilon)\phi_t^1}\left(f(X_0) - f(X_*) + \phi_t^2 \frac{L_0(\overline{\omega} + d(\overline{\sigma_1^2\omega} + \overline{\sigma_1^2})) + L_1 d\left(\overline{\sigma_0^2} + \overline{\sigma_0^2\omega}\right)}{2N}\right) \overset{t\to\infty}{\to} 0. \tag{115}$$

### B.2.2  OUR NEW FIRST-ORDER SDE FOR DCSGD

The following is the *first*-order SDE model of DCSGD and is a straightforward generalization of Theorem 3.6 in Compagnoni et al. (2025a) and Remark A.2. Let us consider the stochastic process $X_t \in \mathbb{R}^d$ defined as the solution of

$$dX_t = -\nabla f(X_t)dt + \frac{\eta}{2}\nabla^2 f(X_t)\nabla f(X_t)dt + \sqrt{\frac{\eta}{N}}\sqrt{\tilde{\Sigma}(X_t)}dW_t, \tag{116}$$

where for $\Phi_{\xi_i, \gamma_i}(x) := \mathcal{C}_{\xi_i}\left(\nabla f_{\gamma_i}(x)\right) - \nabla f_{\gamma_i}(x)$

$$\tilde{\Sigma}(x) = \frac{1}{N}\sum_{i=1}^N \left(\mathbb{E}_{\xi_i\gamma_i}\left[\Phi_{\xi_i,\gamma_i}(x)\Phi_{\xi_i,\gamma_i}(x)^\top\right] + \Sigma_i(x)\right). \tag{117}$$

**Theorem B.5.** *Let $f$ be $(L_0, L_1)$-smooth, the learning rate scheduler $\eta_t$ such that $\phi_t^i = \int_0^t (\eta_s)^i ds$, $\phi_t^1 \overset{t\to\infty}{\to} \infty$, $\frac{\phi_t^2}{\phi_t^1} \overset{t\to\infty}{\to} 0$, and $\overline{\sigma^2\omega} := \frac{1}{N}\sum_{i=1}^N \sigma_i^2\omega_i$. Then, for $0 < \epsilon < 1$,*

$$\eta\eta_t < \frac{2\epsilon}{L_0 + L_1\mathbb{E}\left[\|\nabla f(X_t)\|_2\right] + \frac{\overline{\omega}L_0 + d\left(\overline{\sigma^2} + \overline{\sigma^2\omega}\right)L_1 + \overline{\omega}L_1\mathbb{E}\left[\|\nabla f(X_t)\|_2\right]}{N}}, \tag{118}$$

*and for a random time $\hat{t}$ with distribution $\frac{\eta_t}{\phi_t^1}$, we have that*

$$\mathbb{E}\left[\|\nabla f(X_{\hat{t}})\|_2^2\right] \leq \frac{1}{\phi_t^1(1-\epsilon)}\left(f(X_0) - f(X_*) + \phi_t^2 \frac{\eta(L_0 + L_1)d}{2N}\left(\overline{\sigma^2} + \overline{\sigma^2\omega}\right)\right) \overset{t\to\infty}{\to} 0. \tag{119}$$

*Proof.* Since it holds that

$$\mathbb{E}_{\xi_i,\gamma_i}\|(\mathcal{C}_{\xi_i}\left(\nabla f_{\gamma_i}(x)\right) - \nabla f(x))\|_2^2 \leq \omega_i\|\nabla f(x)\|_2^2 + d\sigma_i^2(\omega_i + 1),$$

we have that for $\mathcal{O}(\text{Noise}) = \sqrt{\tilde{\Sigma}(X_t)}\nabla f(X_t)dW_t,$

$$d(f(X_t) - f(X_*)) = -\eta_t\|\nabla f(X_t)\|_2^2 dt + \frac{\eta\eta_t^2}{2}\left(\nabla f(X_t)\right)^\top \nabla^2 f(X_t)\nabla f(X_t)dt + \mathcal{O}(\text{Noise}) \tag{120}$$

$$+ \frac{\eta\eta_t^2}{2}\frac{(L_0 + L_1\|\nabla f(X_t)\|_2)}{N}\left(\frac{1}{N}\sum_{i=1}^N \mathbb{E}_{\xi_i,\gamma_i}\|(\mathcal{C}_{\xi_i}\left(\nabla f_{\gamma_i}(x)\right) - \nabla f(x))\|_2^2\right)dt \tag{121}$$

$$\leq -\eta_t\|\nabla f(X_t)\|_2^2 dt + \frac{\eta\eta_t^2}{2}(L_0 + L_1\|\nabla f(X_t)\|)\|\nabla f(X_t)\|^2 dt + \mathcal{O}(\text{Noise}) \tag{122}$$

$$+ \frac{\eta\eta_t^2}{2}\frac{(L_0 + L_1\|\nabla f(X_t)\|_2)}{N}\left(\overline{\omega}\|\nabla f(X_t)\|_2^2 + \overline{\sigma^2}d + d\overline{\sigma^2\omega}\right)dt. \tag{123}$$

**Phase 1:** If $\|\nabla f(X_t)\|_2 \le 1$, then we have that

$$d(f(X_t) - f(X_*)) \le \|\nabla f(X_t)\|_2^2 \left( \eta_t - \frac{\eta_t^2 \eta}{2}(L_0 + L_1)\left(1 + \frac{\overline{\omega}}{N}\right) \right) dt \tag{124}$$

$$+ (\eta_t)^2 \frac{\eta(L_0 + L_1)d}{2N}\left(\overline{\sigma^2} + \overline{\sigma^2\omega}\right) dt + \mathcal{O}(\text{Noise}). \tag{125}$$

**Phase 2:** If $\|\nabla f(X_t)\|_2 > 1$, we have that

$$d(f(X_t) - f(X_*)) \le -\eta_t \|\nabla f(X_t)\|_2^2 dt + \frac{\eta \eta_t^2}{2}(L_0 + L_1\|\nabla f(X_t)\|)\|\nabla f(X_t)\|^2 dt + \mathcal{O}(\text{Noise}) \tag{126}$$

$$+ (\eta_t)^2 \frac{\eta(L_0 + L_1\|\nabla f(X_t)\|_2)}{2N}\left(\overline{\omega}\|\nabla f(X_t)\|_2^2 + \overline{\sigma^2}d + d\overline{\sigma^2\omega}\right) dt \tag{127}$$

$$\le -\eta_t \|\nabla f(X_t)\|_2^2 \left[ 1 - \frac{\eta_t \eta}{2}\left[ (L_0 + L_1\|\nabla f(X_t)\|_2)\left[1 + \frac{\overline{\omega}}{N}\right] + \frac{d\left(\overline{\sigma^2} + \overline{\sigma^2\omega}\right)L_1}{N} \right] \right]$$

$$+ \eta_t^2 \frac{\eta L_0 d}{2N}\left(\overline{\sigma^2} + \overline{\sigma^2\omega}\right) + \mathcal{O}(\text{Noise}). \tag{128}$$

By taking a worst-case scenario approach, we merge these two bounds into a single one. With arguments that follow the same steps we detailed in the proof of Theorem B.1, we have that for $0 < \epsilon < 1$, we have that if

$$\eta \eta_t < \frac{2\epsilon}{L_0 + L_1\|\nabla f(X_t)\|_2 + \frac{\overline{\omega}L_0 + d\left(\overline{\sigma^2} + \overline{\sigma^2\omega}\right)L_1 + \overline{\omega}L_1\|\nabla f(X_t)\|_2}{N}}, \tag{129}$$

by integrating over time and by the Law of the Unconscious Statistician, we have that

$$\mathbb{E}\left[\|\nabla f(X_{\hat{t}})\|_2^2\right] \le \frac{1}{\phi_t^1(1 - \epsilon)}\left( f(X_0) - f(X_*) + \phi_t^2 \frac{\eta(L_0 + L_1)d}{2N}\left(\overline{\sigma^2} + \overline{\sigma^2\omega}\right) \right) \overset{t \to \infty}{\to} 0, \tag{130}$$

where $\hat{t}$, is a random time with distribution $\frac{\eta_{\hat{t}}}{\phi_t^1}$.

Finally, for practical reasons, we leverage the distributed setting to tighten the requirements on the learning rate scheduler to make it experimentally viable, and rather require

$$\eta \eta_t < \frac{2\epsilon}{L_0 + L_1 \mathbb{E}\left[\|\nabla f(X_t)\|_2\right] + \frac{\overline{\omega}L_0 + d\left(\overline{\sigma^2} + \overline{\sigma^2\omega}\right)L_1 + \overline{\omega}L_1 \mathbb{E}[\|\nabla f(X_t)\|_2]}{N}}. \tag{131}$$

$$\square$$

Finally, one can generalize this result to cover the $(\sigma_0^2, \sigma_1^2)$-Variance.

**Theorem B.6.** *Let $f$ be $(L_0, L_1)$-smooth, $\max(\Sigma_i(x)) < \sigma_{i,0}^2 + \sigma_{i,1}^2\|\nabla f(x)\|_2^2$, the learning rate scheduler $\eta_t$ such that $\phi_t^i = \int_0^t (\eta_s)^i ds$, $\phi_t^1 \overset{t \to \infty}{\to} \infty$, $\frac{\phi_t^2}{\phi_t^1} \overset{t \to \infty}{\to} 0$, $\overline{\sigma_0^2} := \frac{1}{N}\sum_{i=1}^N \sigma_{0,i}^2$, $\overline{\sigma_1^2} := \frac{1}{N}\sum_{i=1}^N \sigma_{1,i}^2$, $\overline{\sigma_0^2\omega} := \frac{1}{N}\sum_{i=1}^N \sigma_{i,0}^2 \omega_i$, and $\overline{\sigma_1^2\omega} := \frac{1}{N}\sum_{i=1}^N \sigma_{i,1}^2 \omega_i$. Then, for $0 < \epsilon < 1$,*

$$\eta \eta_t < \frac{2\epsilon}{L_0 + L_1 \mathbb{E}\left[\|\nabla f(X_t)\|_2\right] + \frac{L_0\left(\overline{\omega} + d(\overline{\sigma_1^2\omega} + \overline{\sigma_1^2})\right) + L_1 d\left(\overline{\sigma_0^2} + \overline{\sigma_0^2\omega}\right) + L_1\left(\overline{\omega} + d(\overline{\sigma_1^2\omega} + \overline{\sigma_1^2})\right)\mathbb{E}[\|\nabla f(X_t)\|_2]}{N}}, \tag{132}$$

*and for a random time $\hat{t}$ with distribution $\frac{\eta_t}{\phi_t^1}$, we have that*

$$\mathbb{E}\left[\|\nabla f(X_{\hat{t}})\|_2^2\right] \le \frac{1}{(1 - \epsilon)\phi_t^1}\left( f(X_0) - f(X_*) + \phi_t^2 \frac{\eta(L_0 + L_1)d(\overline{\sigma_0^2} + \overline{\sigma_0^2\omega})}{2N} \right) \overset{t \to \infty}{\to} 0. \tag{133}$$

## B.3 DISTRIBUTED SIGNSGD

### B.3.1 FIRST ORDER SDE

The following is the *first*-order SDE model of DSignSGD (see Theorem 3.10 in Compagnoni et al. (2025a)). Let us consider the stochastic process $X_t \in \mathbb{R}^d$ defined as the solution of

$$dX_t = -\frac{1}{N}\sum_{i=1}^N \left(1 - 2\mathbb{P}(\nabla f_{\gamma_i}(X_t) < 0)\right) dt + \sqrt{\frac{\eta}{N}}\sqrt{\overline{\Sigma}(X_t)}dW_t. \tag{134}$$

where

$$\overline{\Sigma}(X_t) := \frac{1}{N} \sum_{i=1}^{N} \overline{\Sigma_i}(X_t), \tag{135}$$

and $\overline{\Sigma_i}(x) = \mathbb{E}[\xi_{\gamma_i}(x)\xi_{\gamma_i}(x)^\top]$ where $\xi_{\gamma_i}(x) := \text{sign}(\nabla f_{\gamma_i}(x)) - 1 + 2\mathbb{P}(\nabla f_{\gamma_i}(x) < 0)$ the noise in the sample sign $(\nabla f_{\gamma_i}(x))$.

**Corollary B.7** (Corollary C.10 in Compagnoni et al. (2025a))**.** *If the stochastic gradients are $\nabla f_{\gamma_i}(x) = \nabla f(x) + \sqrt{\Sigma_i} Z_i$ such that $Z_i \sim t_\nu(0, I_d)$ does not depend on $x$, $\nu$ are the degrees of freedom, and scale matrices $\Sigma_i = \text{diag}(\sigma_{1,i}^2, \cdots, \sigma_{d,i}^2)$. Then, the SDE of DSignSGD is*

$$dX_t = -\frac{2}{N} \sum_{i=1}^{N} \Xi_\nu \left( \Sigma_i^{-\frac{1}{2}} \nabla f(X_t) \right) dt + \sqrt{\frac{\eta}{N}} \sqrt{\tilde{\Sigma}(X_t)} dW_t. \tag{136}$$

*where $\Xi_\nu(x)$ is defined as $\Xi_\nu(x) := x \frac{\Gamma\left(\frac{\nu+1}{2}\right)}{\sqrt{\pi\nu}\Gamma\left(\frac{\nu}{2}\right)} {}_2F_1\left(\frac{1}{2}, \frac{\nu+1}{2}; \frac{3}{2}; -\frac{x^2}{\nu}\right)$, ${}_2F_1(a, b; c; x)$ is the hypergeometric function, and*

$$\tilde{\Sigma}(X_t) := I_d - \frac{4}{N} \sum_{i=1}^{N} \left( \Xi_\nu \left( \Sigma_i^{-\frac{1}{2}} \nabla f(X_t) \right) \right)^2. \tag{137}$$

**Theorem B.8.** *Let $f$ be $(L_0, L_1)$-smooth, $\eta_t$ a learning rate scheduler such that $\phi_t^i = \int_0^t (\eta_s)^i ds$, $\phi_t^1 \overset{t \to \infty}{\to} \infty$, $\frac{\phi_t^2}{\phi_t^1} \overset{t \to \infty}{\to} 0$, $\Sigma_i \leq \sigma_{max,i}^2$, $\sigma_{\mathcal{H},1}$ be the harmonic mean of $\{\sigma_{max,i}\}$, and $\ell_\nu := 2\Xi_\nu'(0) > 0$ a constant. Then, for a scheduler $\eta\eta_t < \frac{2N\ell_\nu}{\sigma_{\mathcal{H},1}dL_1}$ and a random time $\tilde{t}$ with distribution $\frac{\eta_t \ell_\nu \sigma_{\mathcal{H},1}^{-1} - \eta_t^2 \frac{\eta L_1 d}{2N}}{\phi_t^1 \ell_\nu \sigma_{\mathcal{H},1}^{-1} - \phi_t^2 \frac{\eta L_1 d}{2N}}$, we have that*

$$\mathbb{E}\|\nabla f(X_{\tilde{t}})\|_2^2 \leq \frac{1}{\phi_t^1 \ell_\nu \sigma_{\mathcal{H},1}^{-1} - \phi_t^2 \frac{\eta L_1 d}{2N}} \left( f(X_0) - f(X_*) + \frac{\eta(L_0 + L_1)d\phi_t^2}{2N} \right) \overset{t \to \infty}{\to} 0. \tag{138}$$

*Proof.* By Itô Lemma on $f(X_t) - f(X_*)$, we have that for $\mathcal{O}(\text{Noise}) = \sqrt{\overline{\Sigma}(X_t)} \nabla f(X_t) dW_t$,

$$d(f(X_t) - f(X_*)) \leq -\ell_\nu \sigma_{\mathcal{H},1}^{-1} \eta_t \|\nabla f(X_t)\|_2^2 dt + \frac{\eta\eta_t^2 d}{2N} (L_0 + L_1 \|\nabla f(X_t)\|_2) dt + \mathcal{O}(\text{Noise}) \tag{139}$$

**Phase 1:** $\|\nabla f(X_t)\|_2 \leq 1$:

$$d(f(X_t) - f(X_*)) \leq -\ell_\nu \sigma_{\mathcal{H},1}^{-1} \eta_t \|\nabla f(X_t)\|_2^2 dt + \frac{\eta\eta_t^2 d}{2N} (L_0 + L_1) dt + \mathcal{O}(\text{Noise}). \tag{140}$$

**Phase 2:** $\|\nabla f(X_t)\|_2 > 1$:

$$d(f(X_t) - f(X_*)) \leq -\ell_\nu \sigma_{\mathcal{H},1}^{-1} \eta_t \|\nabla f(X_t)\|_2^2 dt + \frac{\eta\eta_t^2 dL_1 \|\nabla f(X_t)\|_2^2}{2N} + \frac{\eta\eta_t^2 dL_0}{2N} dt + \mathcal{O}(\text{Noise}). \tag{141}$$

By taking the worst case of these two phases, we have that

$$d(f(X_t) - f(X_*)) \leq -\ell_\nu \sigma_{\mathcal{H},1}^{-1} \eta_t \|\nabla f(X_t)\|_2^2 dt + \frac{\eta\eta_t^2 dL_1 \|\nabla f(X_t)\|_2^2}{2N} dt + \frac{\eta\eta_t^2 d}{2N} (L_0 + L_1) dt + \mathcal{O}(\text{Noise}). \tag{142}$$

With arguments that follow the same steps we detailed in the proof of Theorem B.1, we have that

$$\mathbb{E}\|\nabla f(X_{\tilde{t}})\|_2^2 \leq \frac{1}{\phi_t^1 \ell_\nu \sigma_{\mathcal{H},1}^{-1} - \phi_t^2 \frac{d\eta L_1}{2N}} \left( f(X_0) - f(X_*) + \frac{\eta(L_0 + L_1)d\phi_t^2}{2N} \right) \overset{t \to \infty}{\to} 0. \tag{143}$$

$\square$

### B.3.2 OUR NEW FIRST-ORDER SDE FOR DSIGNSGD

The following is the *first*-order SDE model of DSignSGD and is a straightforward generalization of Corollary C.10 in Compagnoni et al. (2025a) and Remark A.2. We observe that $\Xi_\nu'(x)$ is bounded by the positive finite constant $M_\nu$.

$$dX_t = -\frac{2}{N} \sum_{i=1}^{N} \Xi_\nu \left( \Sigma_i^{-\frac{1}{2}} \nabla f(X_t) \right) dt$$

$$+ \frac{\eta}{N} \sum_{i=1}^{N} \Sigma_i^{-\frac{1}{2}} \nabla^2 f(X_t) \left( \Xi_\nu' \left( \Sigma_i^{-\frac{1}{2}} \nabla f(X_t) \right) \circ \Xi_\nu \left( \Sigma_i^{-\frac{1}{2}} \nabla f(X_t) \right) \right) dt$$

$$+ \sqrt{\frac{\eta}{N}} \sqrt{\tilde{\Sigma}(X_t)} dW_t. \tag{144}$$

**Theorem B.9.** *Let $f$ be $(L_0, L_1)$-smooth, $\Sigma_i \le \sigma_{max,i}^2$, $\sigma_{\mathcal{H},1}$ be the harmonic mean of $\{\sigma_{max,i}\}$, $M_\nu :=$ $\sup\{\Xi_\nu'(x)\} > 0$ and $\ell_\nu := 2\Xi_\nu'(0) > 0$ constants, and $K := \left( \frac{L_1}{2N} + \frac{(L_0+L_1)\sigma_{\mathcal{H},1}^{-1} M_\nu}{\sqrt{d}} \right)$. Then, for a scheduler $\eta\eta_t < \frac{\ell_\nu K^{-1}}{\sigma_{\mathcal{H},1} d}$ and a random time $\tilde{t}$ with distribution $\frac{\eta_t \ell_\nu \sigma_{\mathcal{H},1}^{-1} - \eta_t^2 K}{\phi_t^1 \ell_\nu \sigma_{\mathcal{H},1}^{-1} - \phi_t^2 K}$, we have that*

$$\mathbb{E}\|\nabla f(X_{\tilde{t}})\|_2^2 \le \frac{1}{\phi_t^1 \ell_\nu \sigma_{\mathcal{H},1}^{-1} - \phi_t^2 K} \left( f(X_0) - f(X_*) + \phi_t^2 \eta(L_0+L_1)d \left( \frac{1}{2N} + \frac{M_\nu}{\sigma_{\mathcal{H},1}\sqrt{d}} \right) \right) \overset{t\to\infty}{\to} 0. \tag{145}$$

*Proof.* By Itô Lemma on $f(X_t) - f(X_*)$, we have that for $\mathcal{O}(\text{Noise}) = \sqrt{\Sigma(X_t)}\nabla f(X_t)dW_t$,

$$d(f(X_t) - f(X_*)) \le -\ell_\nu \sigma_{\mathcal{H},1}^{-1} \eta_t \|\nabla f(X_t)\|_2^2 dt + \eta\eta_t^2 \sigma_{\mathcal{H},1}^{-1}(L_0 + L_1\|\nabla f(X_t)\|_2)M_\nu \|\nabla f(X_t)\|_1 dt \tag{146}$$

$$+ \frac{\eta\eta_t^2 d}{2N}(L_0 + L_1\|\nabla f(X_t)\|_2)dt + \mathcal{O}(\text{Noise}). \tag{147}$$

**Phase 1:** $\|\nabla f(X_t)\|_2 \le 1$:

$$d(f(X_t) - f(X_*)) \le -\ell_\nu \sigma_{\mathcal{H},1}^{-1} \eta_t \|\nabla f(X_t)\|_2^2 dt + \eta\eta_t^2 \sigma_{\mathcal{H},1}^{-1}(L_0 + L_1)M_\nu \sqrt{d}dt \tag{148}$$

$$+ \frac{\eta\eta_t^2 d}{2N}(L_0 + L_1)dt + \mathcal{O}(\text{Noise}). \tag{149}$$

**Phase 2:** $\|\nabla f(X_t)\|_2 > 1$: Since $\|\nabla f(X_t)\|_1 < \sqrt{d}\|\nabla f(X_t)\|_2 < \sqrt{d}\|\nabla f(X_t)\|_2^2$, we have that

$$d(f(X_t) - f(X_*)) \le -\ell_\nu \sigma_{\mathcal{H},1}^{-1} \eta_t \|\nabla f(X_t)\|_2^2 dt + \eta\eta_t^2 \sigma_{\mathcal{H},1}^{-1}(L_0 + L_1)M_\nu \sqrt{d}\|\nabla f(X_t)\|_2^2 dt \tag{150}$$

$$+ \frac{\eta\eta_t^2 dL_1 \|\nabla f(X_t)\|_2^2}{2N} + \frac{\eta\eta_t^2 dL_0}{2N}dt + \mathcal{O}(\text{Noise}). \tag{151}$$

By taking the worst case of these two phases, we have that

$$d(f(X_t) - f(X_*)) \le -\ell_\nu \sigma_{\mathcal{H},1}^{-1} \eta_t \|\nabla f(X_t)\|_2^2 dt + \eta\eta_t^2 \sigma_{\mathcal{H},1}^{-1}(L_0 + L_1)M_\nu \sqrt{d}\|\nabla f(X_t)\|_2^2 dt \tag{152}$$

$$+ \frac{\eta\eta_t^2 dL_1 \|\nabla f(X_t)\|_2^2}{2N}dt + \eta\eta_t^2(L_0 + L_1)d \left( \frac{1}{2N} + \frac{M_\nu}{\sigma_{\mathcal{H},1}\sqrt{d}} \right) dt + \mathcal{O}(\text{Noise}). \tag{153}$$

With arguments that follow the same steps we detailed in the proof of Theorem B.1, we have that

$$\mathbb{E}\|\nabla f(X_{\tilde{t}})\|_2^2 \le \frac{1}{\phi_t^1 \ell_\nu \sigma_{\mathcal{H},1}^{-1} - \phi_t^2 K} \left( f(X_0) - f(X_*) + \phi_t^2 \eta(L_0+L_1)d \left( \frac{1}{2N} + \frac{M_\nu}{\sigma_{\mathcal{H},1}\sqrt{d}} \right) \right) \overset{t\to\infty}{\to} 0. \tag{154}$$

$\square$

## B.4 LIMITATIONS

Our analysis focuses on *homogeneous* client distributions to isolate the effects of noise, compression, and adaptivity without the additional complexity of data heterogeneity. Extending the results to heterogeneous settings—where clients may have different tail indices, variance structures, or asymmetric noise—is an important direction for future work, and our framework is fully compatible with such extensions. We also restrict attention to *unbiased* and *signed* gradient compression, while many practical distributed optimizers employ general *biased* compressors or use *error-feedback* (EF) mechanisms to recover convergence guarantees. Our SDE framework naturally accommodates EF by modifying the drift term to include the memory state and can be extended to biased compressors via suitable bias-correction terms in the continuous-time limit, providing a direct foundation for these future developments.

Additionally, we focus on finite-sum minimization as per the literature (Jastrzebski et al., 2018), and do not tackle questions related to generalization (Smith et al., 2020).

Finally, our contribution is intentionally foundational: Rather than proposing new optimizers, we build a rigorous, unified framework that captures the joint effects of noise, compression, and adaptivity for distributed methods under $(L_0, L_1)$-smoothness. We view this work as a basis for future extensions (e.g., heterogeneous clients, error-feedback, and general biased compressors) and for subsequent analyses that further systematize large-scale stochastic optimization.

**Acknowledgments.** We acknowledge the use of OpenAI's ChatGPT as a writing assistant to help us rephrase and refine parts of the manuscript. All technical content, derivations, and scientific contributions remain the sole responsibility of the authors.

# C EXPERIMENTS

Our experiments are intentionally minimalistic: They are designed to validate the fidelity of the derived insights and to illustrate the qualitative phenomena predicted by our theory, rather than to benchmark performance on specific tasks. This aligns with the theoretical nature of our contribution.

## C.1 DCSGD - FIGURE 1 - (LEFT COLUMN)

We optimize $f(x) = \frac{\sum_{j=1}^{1000}(x_j)^4}{4}$ as we inject Gaussian noise with mean 0 and variance $\sigma^2 \|\nabla f(x)\|_2^2$ on the gradient. The learning rate is $\eta = 0.1$, $\sigma = 0.1$}, we use *random sparsification* with $\omega \in \{4, 8, 16\}$, and we average over 1000 runs. In the top figure, we use no scheduler, while in the bottom one we use a scheduler as per Eq. 16.

## C.2 DSIGNSGD - FIGURE 1 - (RIGHT COLUMN)

We optimize $f(x) = \frac{x^4}{4}$ as we inject student's t noise with $\nu = 1$ and scale parameters $\sigma$ on the gradient. The learning rate is $\eta = 0.1$, $\sigma \in \{0.25, 0.5, 1, 2, 8, 16\}$, and we average over 10000 runs. In the top figure, we use no scheduler, while in the bottom one we use a scheduler as per Theorem 4.3, e.g. $\eta_t = \frac{1}{\sqrt{t+1}}$.

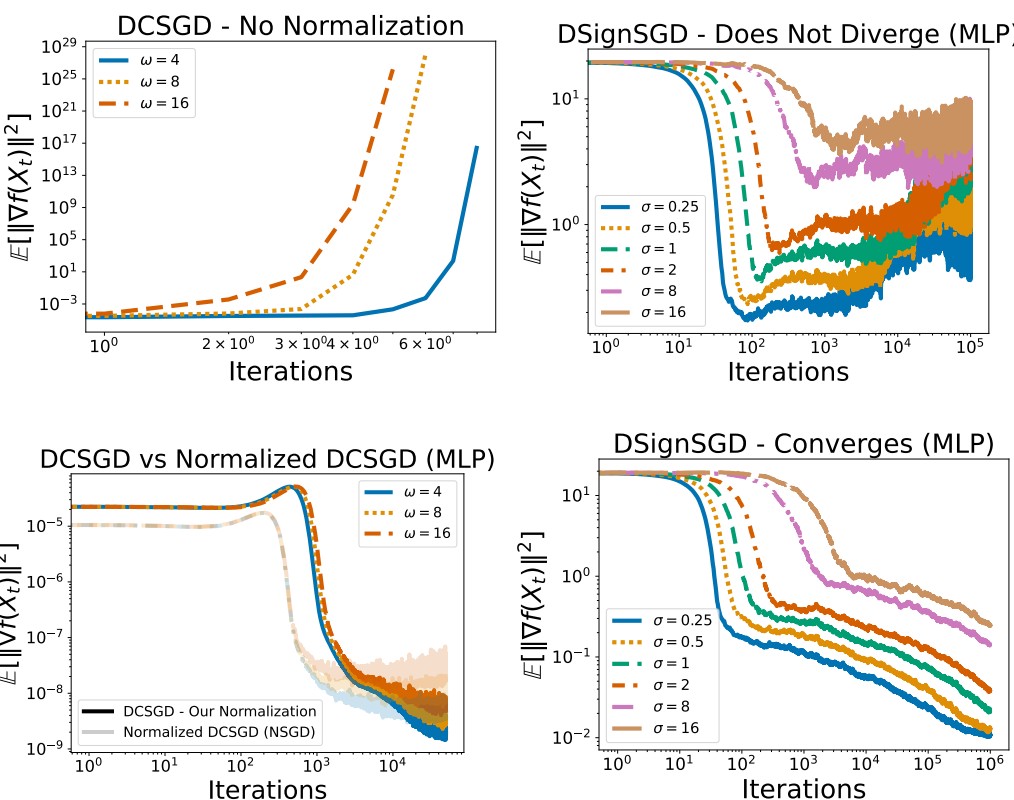

Figure C.1: DSignSGD, Compressed DCSGD, and Normalized SGD on a small MLP regression problem. We train a one-hidden-layer ReLU MLP. Consistently with Theorem 4.2, DCSGD with compressed noisy gradients diverges more quickly as the compression becomes more aggressive when the step-size normalization proposed in Eq. 16 is *not* used, while the normalized DCSGD scheme remains stable and convergent for all compression levels. For comparison, we also report a baseline based on plain normalized SGD with the same gradient noise and compression, which exhibits a different stability profile from our normalized DCSGD, which looks less stable.

## C.3 EXPERIMENTS ON A MLP (FIGURE C.1)

**Architecture, data, and metric.** We consider fully connected neural network $h_\theta : \mathbb{R}^{20} \to \mathbb{R}$ with a single hidden layer of width 64 and ReLU activation. The network has the form

$$h_\theta(x) = W_2 \, \phi(W_1 x + b_1) + b_2, \tag{155}$$

where $W_1 \in \mathbb{R}^{64 \times 20}$, $W_2 \in \mathbb{R}^{1 \times 64}$, $b_1 \in \mathbb{R}^{64}$, $b_2 \in \mathbb{R}$, and $\phi$ denotes the ReLU activation. We collect all parameters into a single vector $\theta \in \mathbb{R}^d$. The loss is the mean squared error

$$f(\theta) = \frac{1}{n} \sum_{i=1}^n \big(h_\theta(x_i) - y_i\big)^2, \tag{156}$$

with a fixed dataset of $n = 4096$ examples. Inputs are $x_i \in \mathbb{R}^{20}$ sampled as $x_i \sim \mathcal{N}(0, I_{20})$, and labels are generated from a linear teacher

$$y_i = x_i^\top w_\star + \varepsilon_i, \qquad w_\star \sim \mathcal{N}(0, I_{20}), \quad \varepsilon_i \sim \mathcal{N}(0, 0.1^2). \tag{157}$$

The dataset $(x_i, y_i)_{i=1}^n$ is sampled once and reused for all methods and repetitions, while the number of clients is $N = 8$. At iteration $t$, we compute the full-batch gradient $g_t = \nabla_\theta f(\theta_t) \in \mathbb{R}^d$ and monitor the quantity

$$\|g_t\|_2^2. \tag{158}$$

For each setting, we approximate the expectation by averaging this quantity over multiple independent runs.

**Noise injection and compression.** In all experiments, we inject additive Gaussian gradient noise and then apply random sparsification. At each iteration $t$, we sample $Z_t \sim \mathcal{N}(0, \sigma^2 \|g_t\|_2^2 . I_d)$ with $\sigma = 0.1$ and form the noisy gradient $g_t + Z_t$. For a given sparsification probability $p \in \{0.8, 0.9, 0.95\}$ we draw an i.i.d. mask $m_t \in \{0, 1\}^d$ with

$$\mathbb{P}\big[(m_t)_i = 1\big] = 1 - p, \tag{159}$$

and define the unbiased random sparsifier

$$C_p(v) = \frac{v \odot m_t}{1 - p}, \tag{160}$$

so that $\mathbb{E}[C_p(v) \mid v] = v$. In the plots, we simply label these three compression levels as $\omega \in \{4, 8, 16\}$, with larger $\omega$ corresponding to more aggressive sparsification (larger $p$).

**DCSGD without scheduler** Here, we use DCSGD with learning rate $\eta = 0.01$, noise level $\sigma = 0.1$, and sparsification probabilities $p \in \{0.8, 0.9, 0.95\}$. For each value of $p$, we run $T_{\text{div}} = 10$ iterations and repeat the experiment over $n_{\text{runs}}^{\text{div}} = 100$ independent initializations of the MLP. We report the average of $\|g_t\|_2^2$ over these runs as a function of the iteration index.

**DCSGD with our scheduler** Here, we use DCSGD with learning rate $\eta = 0.01$ and use an adaptive scheduler $\eta_t$ as per Eq. 16 where $\sigma_0 = 0$, and we assume $L_0 = L_1 = 1$, because these constants are not actually known. As before, we use $\sigma = 0.1$ and $p \in \{0.8, 0.9, 0.95\}$. For each value of $p$, we run $T_{\text{conv}} = 50000$ iterations and repeat the experiment over $n_{\text{runs}}^{\text{conv}} = 5$ independent initializations. We then average $\|g_t\|_2^2$ over these runs.

**Normalized SGD with compression** This experiment provides a baseline where we apply plain Normalized DCSGD. We use learning rate $\eta = 0.01$, noise level $\sigma = 0.1$, sparsification probabilities $p \in \{0.8, 0.9, 0.95\}$, and a small constant $\varepsilon = 10^{-8}$ added for numerical stability. The horizon and number of runs are the same as in the convergent DCSGD experiment, that is $T_{\text{conv}} = 50000$ iterations and $n_{\text{runs}}^{\text{conv}} = 5$ independent initializations for each value of $p$. As before, we track and report the averaged trajectories of $\|g_t\|_2^2$.

**DSignSGD** Here, we apply DSignSGD as we inject Student's t noise with $\nu = 1$ and scale parameters $\sigma$ on the gradient. The learning rate is $\eta = 0.01$, $\sigma \in \{0.25, 0.5, 1, 2, 8, 16\}$, and we average over 5 runs. In the top figure, we use no scheduler, while in the bottom one we use a scheduler as per Theorem 4.3, $\eta_t = \frac{1}{\sqrt{t+1}}$. As before, we track and report the averaged trajectories of $\|g_t\|_2^2$.

## C.4 CONSTRUCTIVE FORM OF THE NORMALIZATION CONDITION

The sufficient conditions for convergence of DCSGD (see Eq. 16) indicate that the learning rate schedule $\eta \eta_t$ should scale inversely with $\mathbb{E}\|\nabla f(X_t)\|$. While this may appear abstract, it admits a natural and practical implementation in the distributed setting.

**Client-side estimation.** At iteration $t$, each client $i$ already computes a stochastic gradient $\nabla f_{i,\gamma_i}(X_t)$ on a local mini-batch $\gamma_i$. We define the local norm estimate as

$$\hat{g}_i^t = \|\nabla f_{i,\gamma_i}(X_t)\|. \tag{161}$$

This requires no additional computation beyond what is standard for mini-batch gradient methods.

**Server-side aggregation.** The server maintains an estimate of the global gradient norm by averaging the client-side estimates as

$$\hat{G}_t = \frac{1}{N}\sum_{i=1}^N \hat{g}_i^t, \tag{162}$$

which provides a consistent approximation of $\mathbb{E}\|\nabla f(X_t)\|$.

**Normalized learning rate.** A learning rate of the form

$$\eta\eta_t \sim \frac{\eta_0}{1+\hat{G}_t} \tag{163}$$

satisfies the normalization condition in our bounds up to stochastic error. This adjustment can be implemented with negligible communication overhead, requiring each client to transmit only a single scalar per iteration.

## D   THEORETICAL FRAMEWORK

In this section, we introduce the theoretical framework, assumptions, and notations used to formally derive the SDE models used in this paper.

**Definition D.1.** Let $G$ denote the set of continuous functions $g : \mathbb{R}^d \to \mathbb{R}$ of at most polynomial growth, namely such that there exist positive integers $k_1, k_2 > 0$ such that $|g(x)| < k_1(1 + \|x\|_2^2)^{k_2}$, for all $x \in \mathbb{R}^d$.

To simplify the notation, we will write

$$b(x+\eta) = b_0(x) + \eta b_1(x) + O(\eta^2),$$

whenever there exists $g \in G$, independent of $\eta$, such that

$$|b(x+\eta) - b_0(x) - \eta b_1(x)| \le g(x)\eta^2.$$

We now introduce the definition of weak approximation, which formalizes in which sense the solution to an SDE, which is a continuous-time random process, models a discrete-time optimizer.

**Definition D.2.** A continuous-time stochastic process $(X_t)_{t\in[0,T]}$ is an $\alpha$-order weak approximation of a discrete stochastic process $(x_k)_{k=0}^{\lfloor T/\eta \rfloor}$ if for every polynomial growth function $g$, there exists a positive constant $C$, independent of $\eta$, such that $\max_{k=0,\dots,\lfloor T/\eta\rfloor} |\mathbb{E}g(x_k) - \mathbb{E}g(X_{k\eta})| \le C\eta^\alpha$. We will often refer to 1-order and 2-order weak approximations as *first-* and *second-*order SDEs.

This framework focuses on approximation in a *weak sense*, meaning in distribution rather than path-wise. Since $G$ contains all polynomials, all the moments of both processes become closer at a rate of $\eta^\alpha$ and thus their distributions. Thus, while the processes exhibit similar average behavior, their sample paths may differ significantly, justifying the term weak approximation.

The key ingredient for deriving the SDE is given by the following result (see Theorem 1, (Li et al., 2017)), which provides sufficient conditions to get a weak approximation in terms of the single step increments of both $X_t$ and $x_k$. Before stating the theorem, we list the regularity assumption under which we are working.

**Assumptions:** Assume that the following conditions are satisfied:

- $f, f_i \in \mathcal{C}_b^8(\mathbb{R}^d, \mathbb{R})$;
- $f, f_i$ and its partial derivatives up to order 7 belong to $G$;
- $\nabla f, \nabla f_i$ satisfy the following Lipschitz condition: there exists $L > 0$ such that

$$\|\nabla f(u) - \nabla f(v)\|_2 + \sum_{i=1}^d \|\nabla f_i(u) - \nabla f_i(v)\|_2 \le L\|u-v\|_2\,;$$

- $\nabla f, \nabla f_i$ satisfy the following growth condition: there exists $M > 0$ such that

$$\|\nabla f(x)\|_2 + \sum_{i=1}^{n} \|\nabla f_i(x)\|_2 \leq M(1 + \|x\|_2).$$

**Lemma D.3.** *Let $0 < \eta < 1$. Consider a stochastic process $X_t, t \geq 0$ satisfying the SDE*

$$dX_t = b(X_t)dt + \sqrt{\eta}\sigma(X_t)dW_t, \qquad X_0 = x \tag{164}$$

*where $b, \sigma$ together with their derivatives belong to $G$. Define the one-step difference $\Delta = X_\eta - x$, and indicate the $i$-th component of $\Delta$ with $\Delta_i$. Then we have*

1. $\mathbb{E}\Delta_i = b_i\eta + \frac{1}{2}\left[\sum_{j=1}^{d} b_j\partial_j b_i\right]\eta^2 + O(\eta^3) \qquad \forall i = 1, \ldots, d;$

2. $\mathbb{E}\Delta_i\Delta_j = \left[b_i b_j + \sigma\sigma_{ij}^{\top}\right]\eta^2 + O(\eta^3) \qquad \forall i, j = 1, \ldots, d;$

3. $\mathbb{E}\prod_{j=1}^{s}\Delta_{i_j} = O(\eta^3) \qquad \forall s \geq 3, \ i_j = 1, \ldots, d.$

*All functions above are evaluated at $x$.*

**Theorem D.4.** *Let $0 < \eta < 1$, $\tau > 0$ and set $T = \lfloor\tau/\eta\rfloor$. Let Assumption D hold and let $X_t$ be a stochastic process as in Lemma D.3. Define $\bar{\Delta} = x_1 - x$ to be the increment of the discrete-time algorithm, and indicate the $i$-th component of $\bar{\Delta}$ with $\bar{\Delta}_i$. If in addition there exist $K_1, K_2, K_3, K_4 \in G$ so that*

1. $|\mathbb{E}\Delta_i - \mathbb{E}\bar{\Delta}_i| \leq K_1(x)\eta^2, \qquad \forall i = 1, \ldots, d;$

2. $|\mathbb{E}\Delta_i\Delta_j - \mathbb{E}\bar{\Delta}_i\bar{\Delta}_j| \leq K_2(x)\eta^2, \qquad \forall i, j = 1, \ldots, d;$

3. $|\mathbb{E}\prod_{j=1}^{s}\Delta_{i_j} - \mathbb{E}\prod_{j=1}^{s}\bar{\Delta}_{i_j}| \leq K_3(x)\eta^2, \qquad \forall s \geq 3, \forall i_j = 1, \ldots, d;$

4. $\mathbb{E}\prod_{j=1}^{s}|\bar{\Delta}_{i_j}| \leq K_4(x)\eta^2, \qquad \forall i_j = 1, \ldots, d.$

*Then, there exists a constant $C$ so that for all $k = 0, 1, \ldots, N$ we have*

$$|\mathbb{E}g(X_{k\eta}) - \mathbb{E}g(x_k)| \leq C\eta. \tag{165}$$

*We say Eq. 164 is an order $1$ weak approximation of the update step of $x_k$.*

