# OpenReview forum: "On the Interaction of Batch Noise, Adaptivity, and Compression, under $(L_0,L_1)$-Smoothness: An SDE Approach"
_ICLR.cc/2026/Conference — Submitted to ICLR 2026_

### Official Review · Reviewer_ahK4 · 2025-10-16

**Soundness:** 3
**Presentation:** 2
**Contribution:** 2
**Rating:** 6
**Confidence:** 3

**Summary:**

The paper proposed new SDE models that enable correct analysis of SGD with proper learning rate restrictions and conditions, under the $(L_0, L_1)$-smoothness condition. The approach was used to proving convergence bound for the models of distributed compressed GD and signed GD, resulting in a more general condition. Other relaxations of assumptions and conclusions are also drawn from this approach.

**Strengths:**

(1) The proposed new SDE approach fixes the previous issue on learning rate restrictions and convergence guarantees under $(L_0, L_1)$-smoothnesses are provided, along with rigorous proofs.

(2) The proposed framework was used to strengthen the convergence guarantee of DCSGD and DSignSGD, under a more general assumption than the commonly used smoothness condition. This also leads to relaxations of certain assumptions originally needed.

(3) Some other insightful conclusions on the interplay of different factors affecting convergence and the convergence of adaptive method under heavy-tailed noise are drawn.

**Weaknesses:**

(1) The main contribution of the paper seems to be the development of a novel and correct SDE framework for analyzing distributed stochastic optimization. It clarifies some subtle inconsistencies in SDE-based approaches, extending them to include compression and noise structure, but it does not introduce new algorithms or fundamentally new convergence phenomena, which makes the contribution rather incremental.

(2) I do agree that the unification of compression, noise and adaptivity are unified for the first time, but each of those components are analyzed before. They are just not presented in a single theorem. I get the sense that combining them is still possible even without the SDE approach, one just needs to solve the complicated algebra correct. Some of the insights seem to be known already.

(3) The empirical evidence is limited even to support the insights.

**Questions:**

See weaknesses.

---

> ### Author Response · Authors · 2025-11-20
> **Many Thanks - Addressed Weaknesses and Questions**
>
> We sincerely thank the Reviewer ahK4 for the constructive and encouraging feedback. In particular, we greatly appreciate the positive remarks highlighting that our **technical contribution** and analysis are backed by **rigorous proofs** and provide **novel** insights into the interaction between noise, compression, and adaptivity under **general assumptions**.
>
> We hope that the following clarifications will help re-evaluate our submission and potentially adjust the score.
>
> ## Weaknesses
>
>
> 1. **"Is your work incremental or foundational?"**
>
>     We appreciate the Reviewer’s observation regarding the nature of our contribution. Our contribution is **foundational**: Indeed, our goal is precisely to **clarify and correct limitations of the classical continuous-time tools** that have been used since the 1990s for analyzing SGD [1]. This literature has been growing steadily in recent years: See Related Works. Concretely, as shown in Section 4.1, for the quadratic loss $f(x) = \frac{\lambda x^2}{2}$ the standard first- and second-order ODE models from the literature (Eqs. (6)–(7)) predict convergence of GD for **all** stepsizes and, in the second-order case, even faster convergence as $\eta$ increases, **thereby missing** the discrete stability condition $\eta < 2/\lambda$. Similarly, as detailed in Appendix A.3.2 for the quartic loss $f(x) = x^4/4$ (Eqs. (42)–(43)), both classical ODEs again predict **unconditional or accelerated convergence**, whereas **our corrected** model matches the discrete-time behavior and captures that GD diverges unless the learning rate scales inversely with the norm of the iterates (Eq. 44). These examples are not cosmetic: they expose a **systematic mismatch** between classical continuous-time surrogates and the true discrete dynamics, and directly motivate our framework.
>    **Rather than** being incremental, we believe our analysis **opens the way** to more accurate and principled SDE-based modeling of optimizers, and we expect that this viewpoint will lead to further developments.
>
>
> 2. **"Some insights were already known & one could work in discrete time"**
>
>     Regarding the concern that the unification of compression, noise, and adaptivity could be achieved without SDEs: while some related insights were indeed derived for **other algorithms** (e.g., Adagrad, Normalized SGD, see Line 107 for references), **the specific insights for Distributed Compressed SGD and Distributed SignSGD under $(L_0,L_1)$-smoothness and affine variance were not previously known**. Table 1 collocates our work w.r.t. the literature: Our setting is the most general.
>    Our results — Theorems 4.2 and 4.3 — provide **new explicit convergence conditions** with interaction terms (between smoothness, compression, variance structure, and heavy-tailedness) that cannot appear in classical first- or second-order SDE approximations, and are not presented in discrete-time analyses yet.
>    While discrete-time derivations might in principle be possible, our corrected continuous-time model yields these results **naturally and transparently** via Itô calculus. This is precisely why continuous-time modeling has been consistently used in optimization for decades [1].
>
> 3. **"Experimental evidence"**
>
>     Concerning the empirical evidence, our paper is **theoretically oriented**, and our results are primarily **stability and learning-rate prescriptions** for DCSGD and DSignSGD under $(L_0,L_1)$-smoothness, affine variance, and heavy-tailed noise, in the same spirit as the classical condition $0 < \eta < 2/L$ for gradient descent. These conditions are not meant to assert that a particular stepsize or normalization will outperform finely tuned baselines on specific benchmarks, but rather to identify **theoretically justified safe regimes** and clarify how geometry, noise, and compression jointly constrain stable operation.
>
>     Consistent with this objective, the synthetic experiments are **diagnostic illustrations** of the theoretical mechanisms (e.g., divergence without normalization for DCSGD, scheduler-dependent behavior of DSignSGD under heavy-tailed noise), rather than large-scale benchmark studies. We intentionally conducted these experiments at a scale that allows **tight control and clear verification** of the predicted behaviors, which is significantly harder to guarantee in large-scale distributed or deep-learning settings. For these reasons, we believe that the current experimental section is appropriate for the theoretical scope of the contribution. Nonetheless, we added a **new experiment**, validating our insights on an **MLP**: The results are **perfectly in line** with our theory. Please, find them in Figure C.1 together with all the implementation details in Appendix C.3.
>
> ## References
>
> [1] Helmke et al., *Optimization and Dynamical Systems*, Springer London (1994).
>
> [2] Compagnoni et al., *Adaptive Methods through the Lens of SDEs: Theoretical Insights on the Role of Noise*, ICLR 2025.

---

### Official Review · Reviewer_cisk · 2025-10-29

**Soundness:** 2
**Presentation:** 3
**Contribution:** 2
**Rating:** 2
**Confidence:** 3

**Summary:**

This paper develops a unified theoretical framework for distributed compressed SGD (DCSGD) and its sign variant distributed SignSGD (DSignSGD) under the $(L_{0},L_{1})$-smoothness condition, to understand the joint effect of gradient noise, communication compression and the use of adaptive update rules. The analysis is based on stochastic differential equations (SDEs), by including higher-order terms,
that allows relatively general gradient noise, including heavy-tailed and affine-variance regimes. Theoretical insights are obtained for DCSGD and DSignSGD that lead to some practical guidance.

**Strengths:**

(1) Even though the SDE approach has been applied before to study optimization algorithms,  no prior work has worked with the $(L_{0},L_{1})$-smoothness condition. Under this condition, convergence bounds are obtained for DCSGD and DSignSGD.
New insights are obtained.

(2) The paper employs higher-order approximation, and claims that both the classic first-order and second-order SDEs from the literature can lead to conclusions that contradict the dynamics of discrete-time SGD.

**Weaknesses:**

(1) The assumption in Theorem 4.1., i.e. equation (4) is hard to check because the right hand side depends on $\mathbb{E}\Vert\nabla f(X_{t})\Vert_{2}$, which is not explicit. If you can provide some tight bounds on $\mathbb{E}\Vert\nabla f(X_{t})\Vert_{2}$, it will be easier to see if (4) can be satisfied.

(2)  The term higher-order SDE in the paper is a bit misleading, because it seems to be a second-order approximation as well, with the only difference being flipping the sign in the drift term in equation (2) to get equation (3). I am not convinced that the so-called higher-order SDE approximation in equation (3) is better than the second-order SDE, i.e. equation (2) from the literature. Let me explain more in detail. The author(s) of the paper first consider ODE approximation for quadratic function and obtain equation (7), which shows faster convergence as $\eta$ increases and claims this is problematic. Actually, I am not sure about the claim. The reason is that for first or second order approximation to hold, $\eta$ has to be small. When $\eta$ is very small, indeed, a larger stepsize will lead to faster convergence. For you to obtain divergence, $\eta$ needs to be large, which is not in the regime of SDE approximation. Then, the author(s) further analyze the ODE dynamics from equation (8) to equation (14) to try to convince why SDE approximation in equation (3) is better than the second-order SDE. However, this is problematic from two perspectives. First of all, SDE is different than ODE.  You are claiming an SDE approximation is better by ignoring the diffusion term. This is problematic, especially because the diffusion term also depends on $\eta$. Second, the first-order and second-order approximation in the literature has theoretical guarantees, that is, the approximation has been proved rigorously, whereas in the current paper, the author(s) simply claim their equation (3) is better, but there is no rigorous approximation result of a particular order. My guess is that the reason the author(s) of the paper reaches a perhaps wrong approximation model to study is because SDE is fundamentally different than ODE. You simply cannot ignore the diffusion term, to argue which ODE approximates better, and then claim which SDE approximates better when both drift and diffusion terms depend on $\eta$. The flip sign from equation (2) to equation (3) might just be because the ignorance of the Brownian noise term and hence the correction term from Ito's calculus. Because of this, the results and insights obtained for DCSGD and DSignSGD lack a rigorous theoretical foundation, and might be problematic.

**Questions:**

(1) In Theorem 4.1., when you define the random time $\hat{t}$, you should mention whether it is independent of the SDE process or not.

(2) In the proof of Theorem B.1., $\mathcal{O}(\text{Noise})$ should be spelled out explicitly as an Ito integral term, and there should be more explanations how you obtain equation (68). Same can be said about the proof of Theorem B.2. and how you obtain (85).

(3) On the right hand side of (67) and (84), you do not have expectations, and hence what on the right hand sides are stochastic. You either missed expectations or there is something wrong here.

---

> ### Author Response · Authors · 2025-11-20
> **Many Thanks - Addressed Weaknesses and Questions**
>
> We sincerely thank the Reviewer cisk for the constructive and encouraging feedback. In particular, we greatly appreciate that you acknowledge that our **novel** use of SDEs under $(L_0,L_1)$-smoothness fills an **important gap in the literature**, that our analysis provides **new insights** into DCSGD and DSignSGD under general noise assumptions, and that our **technical contribution** offer a remedy to the contradictions between classical SDE models and true discrete-time dynamics.
>
> We hope that the following clarifications will help re-evaluate our submission and potentially adjust the score.
>
> ## Weaknesses
>
>  1. **"Dependence on $\mathbb{E}\|\nabla f(X_t)\|_2$?"**
>
>     Thank you for pointing out the need for additional clarity. Eq. (4) in Thm. 4.1 is indeed a sufficient condition; the dependence on $\mathbb{E}\|\nabla f(X_t)\|_2$ reflects that, under $(L_0,L_1)$-smoothness, no *a priori* uniform stepsize can guarantee convergence. Some normalization proportional to the (expected) gradient norm is necessary, just as in the deterministic setting. This is precisely the phenomenon our model is designed to capture, which is otherwise **not captured** by classic models [5]: See Section 4.1 and Section 4.2.
>
>     We emphasize that it is standard for the quantities appearing in stability conditions (such as $L_0$, $L_1$, or variance parameters) to be unknown in practice, even though they canonically arise in theoretical guarantees. As discussed in Appendix C.3, the term $\mathbb{E}\|\nabla f(X_t)\|_2$ can be approximated on the fly using quantities that are already available during training (for instance, empirical mini-batch gradients). However, the dependency on additional latent problem parameters (e.g., $L_0$, $L_1$, or affine-variance coefficients), which are themselves unknown and would also need to be estimated, makes it rather impractical to check the validity of Eq. 4 in a fully explicit, data-dependent way.
>
> 2. We break down our response to this point for clarity:
>
>    **2.a: "Are your models first or a second order? Why are they better than the classic models?"**
>
>     As shown in Theorem A.4 and Theorem A.5, *“flipping the sign”* converts the **second**-order model from the **literature** [5] into a **first**-order model (**ours**). See Lines 903–916 for the full derivation. Therefore, Eq. (3) is **not** a second-order model, but a **first-order** one.
>    Its advantage comes from **reproducing the correct loss dynamics of the discrete algorithm**. Classical models **fail** in this respect (Lines 358–376), whereas our model **succeeds** (Lines 377–409). Additional details are given in Appendix A.3.2. Finally, Section A.4 directly investigates the possibility of **directly modeling the dynamics of the loss** of SGD: Such an SDE is **consistent** with that derived from our newly proposed model, confirming that our models **truly capture the dynamics of the loss**, contrary to models in the literature.
>
>    **2.b "SDEs model the optimizers only when $\eta$ is small, therefore your claims are misleading"**
>
>     A crucial point here is that the issue is **not** simply about using “large” learning rates. As we detail in our Introduction, under $(L_0,L_1)$-smoothness, there may be **no universal stepsize $\eta>0$ that guarantees stability uniformly over all initializations**, even when $\eta$ is **arbitrarily small**. As we show in Appendix A.3.2 for the quartic loss $f(x) = \tfrac{1}{4}x^4$, for every fixed $\eta>0$ (however small) one can choose $|x_0|$ large enough that discrete-time GD becomes unstable, while the **classical first- and second-order ODE/SDE surrogates still predict unconditional convergence and even “accelerated” decay**. In other words, even in the nominal “small stepsize” regime, these models **fail to encode the fact that admissible learning rates must scale with the gradient norm**. Our SDEs are explicitly designed to **capture exactly this phenomenon: it reproduces the absence of a universal fixed stepsize** which we formalize for general $(L_0,L_1)$-smooth functions in Theorem 4.1 and Theorems 4.2–4.3.
>
>     Finally, it appears that the learning rate does **not** need to be small **in practice** for SDEs to track their respective algorithms: We could not find many papers that empirically verify the quality of SDE approximations and predictions, but those that we found use learning rates up to $0.01$ [1,2,3], and [4] uses rates up to $0.1$ even on a GPT-2–style model.

---

> ### Author Response · Authors · 2025-11-20
> **Continuation & References**
>
> ## Weaknesses (Continuation)
>
>
>    **2.c "Are you ignoring diffusion terms in your analyses?"**
>
> **At no point do we discard the diffusion component, nor do we base any argument on an ODE-only approximation.**
>    Regarding the claim that our analysis “ignores the diffusion term,” this appears to stem from a misunderstanding, as the statement is incorrect. Indeed, our analysis throughout the paper explicitly relies on **full SDE models**, including both drift and diffusion terms. These appear, for instance, in Eqs. (53), (70), (87), (106), (124), (126), and (134), where the diffusion structure is carefully derived from the mini-batch noise model under $(L_0,L_1)$-smoothness and affine variance assumptions.
>
>    What we **do** use ODEs for is **purely didactic motivation**: Section 4.2 illustrates that **even in the deterministic setting**, classical continuous-time surrogates already fail to replicate the correct discrete-time stability behaviour under $(L_0,L_1)$-smoothness. This is intentional and **pedagogical**: We want to avoid overloading the discussions with technicalities intrinsic in Itô calculus. By showing that the standard ODE models break down **before noise is even introduced**, we motivate why a corrected SDE model is needed in the first place. Importantly, this **ODE discussion is not used for any theoretical claim** in the stochastic setting; it merely highlights, in the simplest possible case, where the classical approximations go wrong: We make this clearer in our revision.
>
>    Finally, we would like to ask the Reviewer what prompted the impression that we “ignore the diffusion term,” since we believe it is important to correct any part of the manuscript that may have been inadvertently misleading. We will be happy to revise the exposition to make our reliance on full SDE models and the purely motivational role of the ODE discussion even clearer.
>
>    **2.d "Do your models enjoy any formal guarantees?"**
>
> Yes, we **formalize** the “sign flip” and **prove in Theorem A.4** that it yields a **first-order model**. Theorem A.5 extends the same reasoning to the SDE. We added a detailed proof in the revision.
>
>    For these reasons, we respectfully but **strongly disagree** with the Reviewer’s claims on this point and invite further discussion to resolve any remaining misunderstandings.
>
> ## Questions
>
> 1. **"Is the random time $\hat{t}$ independent of the SDE process?"**
>
>     Yes. $\hat{t}$ is drawn with density $\eta_t / \phi^1_t$, determined solely by the (deterministic) scheduler. It is therefore independent of the filtration generated by $(X_t, W_t)$. We will state this explicitly below Eq. (5): It is common to give guarantees for a point that is sampled from the history [6, 7].
>
> 2. **Can the missing intermediate steps for Eqs. (68) and (85) be made explicit?**
>
>     We added all missing details in the revised version. For Eqs. (68) and (85), the intermediate steps are straightforward, and we were happy to include the full derivations for completeness. Thank you for pointing this out.
>
> 3. **Can the expressions in Eqs. (67) and (84) be clarified regarding expectations?**
>
>     We added more precise calculations in the revision to make explicit why the current expressions are correct. We apologize for the lack of clarity.
>
> ## References
>
> [1] Compagnoni et al. *An SDE for Modeling SAM: Theory and Insights*, ICML 2023
>
> [2] Compagnoni et al. *SDEs for Minimax Optimization*, AISTATS 2024
>
> [3] Compagnoni et al. *Adaptive Methods through the Lens of SDEs: Theoretical Insights on the Role of Noise*, ICLR 2025
>
> [4] Compagnoni et al. *Unbiased and Sign Compression in Distributed Learning: Comparing Noise Resilience via SDEs*, AISTATS 2025 (**Oral**)
>
> [5] Li et al. *Stochastic Modified Equations and Adaptive Stochastic Gradient Algorithms*, ICML 2018
>
> [6] Stich et al. *On communication compression for distributed optimization on heterogeneous data*, arXiv preprint arXiv:2009.02388 (2020)
>
> [7] Stich et al. *The error-feedback framework: Better rates for SGD with delayed gradients and compressed communication*, JMLR (2020)

---

### Official Review · Reviewer_zQGA · 2025-10-30

**Soundness:** 4
**Presentation:** 4
**Contribution:** 3
**Rating:** 8
**Confidence:** 2

**Summary:**

This paper presents a detailed theoretical study on distributed stochastic optimization by modeling the dynamics of compressed and adaptive gradient methods through stochastic differential equations (SDEs) under the recently introduced (L_0,L_1)-smoothness framework. The authors analyze two representative distributed methods, Distributed Compressed SGD (DCSGD) and Distributed SignSGD (DSignSGD), and highlight the limitations of conventional first- and second-order SDE approximations. They propose refined SDE models that correctly capture learning rate constraints, stability thresholds, and the interaction between gradient noise, compression, and adaptivity. The theoretical results reveal that DSignSGD remains stable even under heavy-tailed noise, while DCSGD requires a normalization factor determined jointly by the compression rate, noise variance, and smoothness parameters. The analysis is complemented by a few illustrative synthetic experiments confirming the theoretical predictions.

**Strengths:**

The paper provides a rigorous and well-developed theoretical exploration of distributed stochastic learning dynamics through SDE analysis. Its contribution lies in unifying the understanding of batch noise, adaptivity, and compression within a realistic smoothness regime that extends beyond classical Lipschitz assumptions.
It provides an interesting integration of SDE-based modeling with distributed optimization theory, offering insights into how continuous-time approximations can reveal stability and convergence properties of discrete stochastic methods. The connection between theoretical results and empirical observations (although limited in scope) is clearly articulated and helps bridge formal analysis with intuition. The manuscript is technically strong, with a clear structure, consistent notation, and effective conceptual explanations that accompany the mathematical derivations

**Weaknesses:**

Despite its depth and rigor, the paper lacks substantial empirical support for the theoretical framework. The presented numerical demonstrations are illustrative but minimal, and the claims regarding practical guidance (e.g., normalization requirements, robustness to heavy-tailed noise) would be more convincing if validated on standard distributed-learning benchmarks.
Furthermore, the practical implications of the analysis (for example, how the derived stability conditions can inform hyperparameter tuning or algorithmic design choices) are only briefly mentioned. A clearer discussion on how practitioners can use the theoretical findings to guide the configuration of distributed systems would strengthen the impact.
As a minor point, the title could more explicitly reflect that the focus is on distributed learning

**Questions:**

Can the proposed analysis be shown to recover or specialize existing known results in simpler settings, such as the single-user (non-distributed) case, or  the uncompressed setting?
Clarifying this would help readers understand how the framework generalizes and extends classical convergence analyses.

Are there examples or empirical indications that the normalization prescriptions derived from the analysis lead to measurable improvements in practical distributed systems?

---

> ### Author Response · Authors · 2025-11-20
> **Many Thanks - Addressed Weaknesses and Questions**
>
> We sincerely thank the Reviewer zQGA for the constructive and encouraging feedback. In particular, we greatly appreciate the positive remarks emphasizing that our **detailed and rigorous theoretical study** offers a **connection between theoretical results and empirical observations is clearly articulated and helps bridge formal analysis with intuition**, and that **the manuscript is technically strong, with a clear structure, consistent notation, and effective conceptual explanations accompanying the mathematical derivations**.
>
> We hope that the following clarifications will help re-evaluate our submission and potentially adjust the score.
>
> ## Weaknesses
>
> 1. **"Experiments"**
>
>     We appreciate that you clearly recognized the focus of our paper: isolating the geometry–noise–compression–adaptivity interaction under $(L_0,L_1)$-smoothness. Much of the contribution is indeed **foundational**, highlighting limitations in existing continuous-time models, which we discuss in Section 4.2 and analyze further in Appendix A.3.2.
>
>     As noted in Appendix C, our experiments are representative: they **validate the fidelity of the theoretical insights** and illustrate the qualitative phenomena predicted by our SDEs, rather than benchmark task-specific performance. We intentionally conducted our experiments that allow **tight control** over all parameters and clearer verification of the theoretical predictions, which is significantly harder to guarantee in large-scale distributed or deep-learning settings. Nonetheless, we added a **new experiment**, validating our insights on an **MLP**: The results are **perfectly in line** with our theory. Please, find them in Figure C.1 together with all the implementation details in Appendix C.3.
>
>
> 2. **"Practitioner Guidance"**
>
>     Thank you for pointing out the need for more explicit practical guidance. First, it is standard for problem-related quantities to appear in step-size restrictions [1]. Second, while many "ingredients" appearing in our normalization are typically **unknown in practice**, our results still offer clear **qualitative** direction. For instance, the derived stability conditions indicate that **stronger compression, heavier-tailed noise, or larger variance should be paired with smaller step sizes**. As with many theoretically derived prescriptions, these insights **do not provide** exact hyperparameter values, as many "ingredients" are not actually known [1], e.g., $L_0$, $L_1$, $\sigma_0$, and so on. Nevertheless, they do offer **useful and principled guidance**. We will clarify this point and expand the discussion on how practitioners can leverage these insights when configuring distributed optimizers.
>
> 3. **"Title Suggestion"**
>
>     Thank you for the suggestion regarding the title. We are happy to adopt the following revised version:
>    **“Distributed Learning under $(L_0,L_1)$-Smoothness: An SDE Approach to the Interaction of Batch Noise, Adaptivity, and Compression.”**
>
> ## Questions
>
> 1. **"Can you recover known results?"**
>
>     Yes. As detailed around Lines 432–439, our analysis recovers known results in simpler settings such as the single-worker or uncompressed limits. For example, our corrected SDE for SGD reproduces the classical stability thresholds, i.e. $\eta < \frac{2}{L}$, while remediating inaccuracies introduced by classical first- **and** second-order SDE approximations.
>
> 2. **Do your step sizes improve performance?**
>
>     Our results are, by design, **stability and learning-rate prescriptions**, in the same spirit as the classical condition $0 < \eta < 2/L$ for gradient descent on an $L$-smooth function. The role of such conditions is to identify **theoretically justified safe regimes** in which divergence is ruled out and the qualitative dynamics are well understood. They are not meant to guarantee that a particular choice of $\eta$ or normalization will outperform all other tuned hyperparameters on a given benchmark. Our contribution is to derive **principled step-size and normalization constraints** that correctly reflect the interaction of $(L_0,L_1)$-smoothness, noise structure, and compression, and to show that these constraints match the behavior observed in controlled experiments. They are not meant to guarantee that a particular choice of $\eta$ or normalization will outperform all other tuned hyperparameters on a given benchmark. Interestingly, [2] shows that SGD with clipping leads to provably faster convergence than constant-stepsize SGD.
>
> ## References
>
> [1] Gorbunov et al., Methods for Convex (L_0,L_1)-Smooth Optimization: Clipping, Acceleration, and Adaptivity, ICLR 2025
>
> [2] Zhang, Jingzhao, et al., Why gradient clipping accelerates training: A theoretical justification for adaptivity, ICLR 2020

---

### Official Review · Reviewer_9dpY · 2025-10-31

**Soundness:** 2
**Presentation:** 2
**Contribution:** 2
**Rating:** 4
**Confidence:** 2

**Summary:**

The paper focuses on non-convex finite sum under the more general $(L_0,L_1)$-smoothness assumption, which relaxes the standard $L$-smoothness condition.  It studies continuous-time analysis via SDE modeling (instead of the standard discrete iteration analysis). Under this setup, the paper analyzes two types of gradient compressors: one is a class of unbiased compressors, and the second is sign-SGD, under assumptions of affine variance/heavy-tailed noise. It argues that standard first and second-order SDE analyses do not accurately reflect the constraints on the learning rate as appear in the standard discrete analysis, and therefore proposes a new second-order SDE model to address this. The paper is primarily theoretical and includes a toy example.

**Strengths:**

* The paper presents a new SGD model that better reflects the results of discrete analysis under a more general $(L_0,L_1)$-smoothness condition and for affine variance/ heavy-tailed noise.
* It provides results for two types of compressors, including Sign-SGD, which is a very practical and efficient method.

**Weaknesses:**

* The motivation for moving to a continuous model (instead of a discrete one) could be strengthened, as the authors argue that the classical continuous model is less accurate than the discrete one; therefore, it should be clarified why we prefer to use SDE over the standard discrete analysis in this case.
* In Theorem 4.3, the term $S_0$ depends on the stochastic variance, which is usually supposed to be independent of it, as in Theorem 4.2. Why does it hold in this case?
* $M_\nu$ and $\mathcal{l}_\nu$ should be briefly defined in the main paper for readability and not only in the appendix, since they are still a part of the main results of the paper. Could the authors provide a brief explanation of each of them?
* The proposed model is limited to second-order SDEs, and it’s unclear how to address the inaccuracy of the standard SDE in first-order modeling, which can be better suited for practical non-convex functions.
* The analysis is limited to finite-sum problems (i.e., guaranteeing minimization of the empirical risk and not generalization), and to a class of unbiased compressors (whereas biased compressors are more common in practice, excluding sign-SGD).

**Questions:**

Please see the weaknesses.

---

> ### Author Response · Authors · 2025-11-20
> **Many Thanks - Addressed Weaknesses and Questions**
>
> We sincerely thank the Reviewer 9dpY for the constructive feedback, and in particular for highlighting that our paper presents: **i)** **novel** SDE models that better reflect discrete-time behavior under **ii)** **more general** setup than the literature $(L_0, L_1)$-smoothness, affine variance/ heavy-tailed noise, and distributed setting), and for appreciating our results for both unbiased compressors and SignSGD, which is a **iii)** **very practical** and **efficient** method.
>
> We hope that the following clarifications will help re-evaluate our submission and potentially adjust the score.
>
> ## Weaknesses
>
> 1. **"Why use a continuous-time SDE model instead of a discrete analysis?"**
>
>     This research direction dates back to the 1990s [1] and has continued to grow steadily since then (see our Related Works). The continuous-time viewpoint gives access to sophisticated tools such as Itô calculus, which have enabled many insightful results (see Lines 102–150). Continuous analyses have historically led to:
>    i) **new conceptual insights**, for example, in [2], where the gradient-flow limit clarified the structure of Nesterov acceleration;
>    ii) **practical insights**, for instance, in [3], where a scaling law relating the learning rate and the batch size was formalized.
>    Moreover, continuous-time methods are naturally well-suited to handle a broad range of noise models, e.g., Gaussian, affine, or even heavy-tailed.
>
>    We follow this literature, highlighting several fundamental issues in the **classical** first- **and** second-order SDE models, and address them accordingly; see Section 4.2 for details. We will clarify this motivation and emphasize that **this approach is popular and well-established** in the literature.
>
> 2. **"Does the term $S_0$ in Theorem 4.3 depend on the stochastic variance?"**
>
>     We apologize, but we do not fully understand this question. In Theorem 4.3, $S_0 := f(X_0) - f(X_*)$ does **not** depend on the stochastic variance. It is simply the initial suboptimality. We kindly ask the Reviewer to specify this concern more precisely so that we can address the misunderstanding.
>
> 3. **"Can the definitions of $M_\nu$ and $\mathcal{l}_\nu$ be made explicit in the main paper?"**
>
>     Of course. $\ell_\nu$ is the slope of the Student's t CDF at the origin and measures how strongly the expected sign update aligns with the true gradient when the gradient is small. $M_\nu$ measures the strongest possible response of the expected sign update to any gradient value, which sets the worst-case stability constraint on the step size. Here are the definitions:
>  $M_{\nu} := \sup \{\Xi_{\nu}^{'}(x)\}>0 $
>  and $ \ell_{\mathbf{\nu}} := 2 \Xi_{\nu}^{'}(0) > 0 $
> , where
> $\Xi_{\nu}(x)$
> is defined as
>  $\Xi_{\nu}(x) := x \frac{\Gamma\left(\frac{\nu+1}{2}\right)}{\sqrt{\pi \nu} \Gamma\left(\frac{\nu}{2}\right)}{ }_2 F_1\left(\frac{1}{2}, \frac{\nu+1}{2} ; \frac{3}{2} ;-\frac{x^2}{\nu}\right)$
>
> and the hypergeometric function is ${}_2 F_1\left(a, b;c; x\right)$ .
>
>
> 4. **"Do your models only address the limitations of the second-order model? Can you handle nonconvex losses?"**
>
>     Our **new** SDEs are **first**-order models that offer an **alternative to both** classical *first*- **and** *second*-order SDEs: they aim at addressing some deficiencies related to the inability of classical models to capture restrictions on the learning rate; the analysis and guarantees do **not rely on convexity** (see Theorems 4.2–4.3): It relies on $(L_0,L_1)$-smoothness. The quadratic and quartic examples in Section 4.2 and Appendix A.3 are purely **pedagogical** and chosen to transparently show where **classical models fail** and where **our corrected models recover the correct** discrete-time stability. We will add a short remark emphasizing that the results apply to general nonconvex losses.

---

> ### Author Response · Authors · 2025-11-20
> **Continuation & References**
>
> ## Weaknesses (Continuation)
>
> 5. **"Limitations to finite-sum problems and unbiased compressors (apart from DPSignSGD)"**
>
>     We agree, and provide our justification below:
> - 5.a We restrict our analysis to finite-sum problems, which is the **standard setting in prior work** (see [3] and our Related Works) and enables sharp and interpretable guarantees: Our goal here is foundational — to isolate the interaction between geometry, noise structure, compression, and adaptivity under $(L_0,L_1)$-smoothness as we put forward a new way to include geometric information into continuous-time models.
> - 5.b While we focus on unbiased compressors, this class is well-studied and central in distributed optimization. For biased compressors, we deliberately analyze Distributed SignSGD, given its communication efficiency and robustness to heavy-tailed noise — properties that are not guaranteed for other biased methods such as Top-$K$. As we discuss in our Conclusion, extending our framework to additional biased compressors (e.g., with error feedback) is an important direction for future work.
> - 5.c We are aware that other SDE-based works have also explored generalization [4,5,6], but this goes beyond the scope of this paper.
>
>     Therefore, we believe **our contribution is solid** in light of the technical contribution, and the combination of general noise and regularity assumptions we tackle. To our knowledge, such a combination has not been addressed before; see Table 1 for a comparison with the literature. We will expand the Limitation section to reflect this discussion.
>
> Finally, we highlight that we added a **new experiment**, validating our insights on an **MLP**: The results are **perfectly in line** with our theory. Please, find them in Figure C.1 together with all the implementation details in Appendix C.3.
>
> ## References
>
> [1] Helmke et al., *Optimization and Dynamical Systems*, Springer London 1994.
>
> [2] Su et al., A differential equation for modeling Nesterov’s accelerated gradient method: Theory and insights. NeurIPS, 2014.
>
> [3] Jastrzębski et al, Three Factors Influencing Minima in SGD, ICANN 2018
>
> [4] Smith et al, On the Generalization Benefit of Noise in Stochastic Gradient Descent, ICML 2020
>
> [5] Smith et al, A Bayesian Perspective on Generalization and Stochastic Gradient Descent, ICLR 2018
>
> [6] Cheng et al, Stochastic Gradient and Langevin Processes, ICML 2020

---

### Official Review · Reviewer_rkAo · 2025-10-31

**Soundness:** 2
**Presentation:** 2
**Contribution:** 2
**Rating:** 2
**Confidence:** 3

**Summary:**

The paper proposes a "corrected" continuous-time model for DCSGD and DSignSGD by flipping the sign of the $\\frac{\\eta}{2}\\nabla^2f\\nabla f$ term in the classic second-order SDE approximation so that the continuous model reproduces the stability threshold of the step size in the GD. It then derives error bounds for a few settings ($(L_0-L_1)$ smoothness, unbiased compression, and affine variance) for both DCSGD and DSignSGD, and shows toy experiments.

**Strengths:**

1. The paper states assumptions (e.g., $L_0-L_1$-smoothness, unbiased compression, affined variance) well with sufficient amount of related works.
2. Proofs are straightforward and mostly consistent with stated assumptions (although the explanation is via toy examples).
3. Overall structure is readable.

**Weaknesses:**

1. Theoretical performance is limited to the error bound of the proposed new SDE approximation only.
2. The justification for relating the SDE’s approximation to the step-size condition in its error bound is weak. I do not understand why a better approximation should also contain the step size in its performance analysis.
3. Normalization claim in Fig. 1 is under-specified and unproven. The figure asserts “normalization stabilizes,” but the algorithm is not specified, and no theorem explains the performance of this normalization mechanism.
4. Some technical proofs are not self-contained.
5. Toy quartics are insufficient to justify claims about stability and training dynamics.

**Questions:**

1. In Line 918, the authors state that
> The following theorem formalizes that our new SDE model from Eq. 27 is formally a first-order weak
approximation for SGD: Its proof is a trivial combination of the arguments in the previous theorem,
and Lemma 1 and Lemma 2 in (Li et al., 2017).

In Eq. (70) of Appendix B.1.2, the authors describe the proposed SDE approximation as a second-order model, which contradicts with their above characterization. A careful, step-by-step justification is needed to show that the approximation satisfies Definition 3.4 (Lines 225–228) with $\\alpha = 1$ or $2$.

Continue the question, if the authors can only show that the proposed SDE is a first-order model, can the authors explain why a sign flip downgrades the model from second order to first order?

2. What is the technical novelty in authors' analysis compared to previous works? It is not specified in the manuscript.

3. The authors heavily cited the paper (Li et al. 2017). The cited paper explained several SGD variants such as momentum SGD and Nesterov accelerated gradient. Could the authors use their proposed SDE for these SGD variants as well?

---

> ### Author Response · Authors · 2025-11-20
> **Many Thanks - Addressed Weaknesses and Questions**
>
> We sincerely thank Reviewer rkAo for the constructive feedback, and for noting the **clear structure**, **straightforward proofs**, and **correct positioning** in the literature.
>
> We hope that the following clarifications will help re-evaluate our submission and potentially adjust the score.
>
> ## Weaknesses
>
> 1. **"Are your formal guarantees derived for the SDE (and not for the discrete algorithm)?"**
>
>     Yes, our formal guarantees are stated at the level of the **continuous-time models** rather than directly for the discrete-time algorithms. This is deliberate: the paper aims to construct and analyze a **faithful SDE surrogates** that correctly encode the interaction of $(L_0,L_1)$-smoothness, affine variance, and compression, together with the resulting step-size restrictions. This modeling step is exactly what the literature on (stochastic) modified equations focuses on: See [1,2,3,4,5,6] for recent, and foundational and influential papers.
>
>     In our case, this does not reduce to an error bound: Our work highlights an **intrinsic limitation** of classic SDEs from the literature. SDEs from the literature fail to capture the learning rate restrictions to ensure the convergence of (for example) SGD. We propose new SDEs that address this limitation and obtain **new stability and normalization prescriptions** (Theorems 4.2 and 4.3, Table 2), which could not be derived from classical first- and second-order SDEs. We will clarify this scope in the introduction and discussion: our contribution is to provide and analyze a continuous-time model that faithfully captures the step-size behavior of the underlying algorithms, rather than to derive separate discrete-time convergence bounds on top of it. Finally:
> - 1.a This theoretical framework **guarantees** that the discrete-time optimizer and the continuous-time model **stay close to each other** up to an error of order $\eta$: See Definition 3.4 for how the notion of "closeness" is formalized.
> - 1.b It has been largely **experimentally** shown that SDEs do track their respective optimizers in several recent works [1,2,5,6], even on CNNs and ViTs.
>
>
> 2.  **"Why should a better SDE approximation also encode step-size restrictions?"**
>
>     As in many areas of science (for instance, physics), a **good model** must reproduce the key aspects of the phenomenon under study. When using SDEs in optimization, we aim to model the dynamics of an optimizer, and there is no doubt that its stability is a **crucial aspect** of these dynamics. Therefore, we argue that any continuous-time approximation that fails to reproduce known stability thresholds cannot be considered faithful for the task at hand.
>
>     The SDE derived for SGD in [6], as well as the classic Gradient Flow [7], are first-order approximations, in the sense that the error between the continuous-time dynamics and the discrete-time dynamics scales with the learning rate $\eta$ (see Definition 3.4 with $\alpha = 1$). A second-order approximation requires adding the term $\frac{\eta}{2} \nabla^2 f(x)\nabla f(x)$ so that the approximation error scales with $\eta^2$. As we discuss on Lines 64–79, these two classic models **both fail** to capture the **step-size restrictions** already known in the literature; Section 4.1 (Lines 329–333) and Section 4.2 detail these phenomena.
>
>     Intuitively, this means that *higher-order SDEs do not necessarily capture the subtleties of discrete-time dynamics any better*. Therefore, we propose new **first-order** models that do recover such **step-size restrictions**. To conclude, it is important to **correctly** encode the step-size in the SDE to ensure that this key feature is preserved. **Failing** to do so leads to **incorrect conclusions when using a continuous-time approximation**. We hope this discussion addresses the Reviewer's concern, and we are happy to answer further questions, as this is one of the key contributions of our work.
>
> 3. **"What normalization is used in Figure 1, and how is it justified?"**
>
>     The caption of Figure 1 specifies **exactly** which normalization has been applied to each algorithm:
>    - **DCSGD**: We use the normalization specified in Theorem 4.2 (see Equation 16).
>    - **DSignSGD**: We use a scheduler that satisfies the assumptions of Theorem 4.3.
>
>     Full experimental details, including the explicit normalization condition and its implementation, are provided in **Appendix C.3**. We will add a link to this section in the revised manuscript to improve traceability.
>
> 4. **"Some proofs are not self-contained"**
>
>     We have provided the fully detailed proof of Theorem A.5 and added supporting technical results in Appendix C. We kindly ask the Reviewer to point us precisely to which other proofs are not self-contained.

---

> ### Author Response · Authors · 2025-11-20
> **Continuation & References**
>
> ## Weaknesses (Continuation)
> 5. **"Insufficient experiments"**
>
>     **We respectfully disagree that these experiments are insufficient.** On the contrary, we believe that the quartic example is extremely relevant for our paper. As fully detailed in Appendix A.3.2, this is an $(L_0, L_1)$-smooth function for which **no universal fixed learning rate ensures convergence**, and this is not captured by ODEs and SDEs from the literature (see Lines 1127-1135), while ours captures this key phenomenon (see Lines 1135-1140). Finally, the purpose of these visual experiments is illustrative - to qualitatively validate the theoretical predictions, not to provide empirical benchmarks. As ours is a theoretical paper, extensive experimentation is beyond its intended scope. Nonetheless, we added a **new experiment**, validating our insights on an **MLP**: The results are **perfectly in line** with our theory. Please, find them in Figure C.1 together with all the implementation details in Appendix C.3.
>
> ## Questions
>
> **1. “Typo & clarification around first and second order”**
>
> We split the answer into subpoints for clarity:
>
> - **1.a** We thank the reviewer for catching this inconsistency. The reference to a “second-order model” in Eq. (70) (Appendix B.1.2) comes from an earlier internal naming convention and is indeed a **typo**. In the revised version, we **consistently** refer to our modified SDE as a **first-order weak approximation** and remove the “second-order” wording there. We regret this oversight and sincerely apologize for the confusion.
>
> - **1.b** In the sense of Definition 3.4, our modified model satisfies the weak-error bound with $\alpha = 1$, as detailed in Theorem A.4 and Theorem A.5. We will make this explicit in the text instead of relying on indirect references. Concretely, Theorem A.4 and A.5 (Lines 908–912) show that **keeping the minus sign** in the drift recovers the classical second-order weak model (order $\alpha = 2$), whereas Lines 913–917 show that **flipping the sign** to a plus yields only first-order weak approximation ($\alpha = 1$). As discussed in our original submission, our construction therefore **intentionally sacrifices the second-order**, enjoyed by the standard second-order SDE of Li et al. [6], in order to obtain a model whose stability region and step-size thresholds match those of the underlying discrete-time algorithm.
>
> 2. **"Technical novelty compared to previous works?"**
>
> Our technical contributions are **fundamental**:
> - 2.a We show how classic **first** and **second**-order models from the literature ([6]) **fail** to capture the **step-size restrictions** in relation to the smoothness of the loss function, e.g., $\eta < \frac{2}{L}$ for GD on an $L$-smooth function. As detailed in Section 4.2, we exemplify these limitations: see Line 347 to Line 357. We introduce a **new first**-order model which addresses these issues: see Lines 377 to 405 --- Appendix A.3 deepens the discussion. Additionally, our **technical contribution opens up the possibility of better modeling of optimizers for the community using SDEs/ODEs in Optimization**, which has grown consistently since the nineties.
> - 2.b To our knowledge, our paper presents the **first SDE analysis under $(L_0, L_1)$-smoothness** that simultaneously handles unbiased compression and affine variance (and, for DSignSGD, heavy-tailed noise) in a distributed setting. Prior SDE papers focus on $L$-smoothness or fixed-covariance models; Table 1 and Table 2 position these results against the literature.
>
> 3. **"Is it possible to extend to Momentum SGD and NAG?"**
> Yes, this is a good comment: We agree that this could be a natural step forward. This work is the first step to harness the potential of **correctly** including second-order information, e.g., the Hessian of the loss, in SDEs to derive meaningful **step-size prescriptions**. We believe our work paves the way to more accurate modeling of optimizers via SDEs, and anticipate that further developments along this direction will yield even deeper insights into the dynamics of modern stochastic optimizers.
>
> ## References
>
> [1] Compagnoni et al. *An SDE for Modeling SAM: Theory and Insights*, ICML 2023
>
> [2] Compagnoni et al. *Unbiased and Sign Compression in Distributed Learning: Comparing Noise Resilience via SDEs*, AISTATS 2025 (**Oral**)
>
> [3] Jastrzębski et al. *Three Factors Influencing Minima in SGD*, CANN 2018
>
> [4] Xiao et al. *Exact Risk Curves of SignSGD in High Dimensions*, ICML 2025
>
> [5] Marshall et al. *To Clip or Not to Clip: The Dynamics of SGD with Gradient Clipping in High Dimensions*, ICLR 2025
>
> [6] Li et al. *Stochastic Modified Equations and Adaptive Stochastic Gradient Algorithms*, ICLR 2018
>
> [7] Standard gradient flow for GD

---

### Author Response · Authors · 2025-11-20
**General Answer**

Dear Reviewers and AC,

We sincerely appreciate your time, thorough reviews, insightful comments, and interesting questions regarding our paper. Your feedback has greatly contributed to the finalization of our work.

We are pleased that **all** Reviewers understood the main **contributions** of this **theoretical** paper:

1. **Fundamental**: We identify **limitations** of existing first- and second- SDE models from the literature: They **fail to capture learning rate restrictions** under $(L_0,L_1)$-smoothness, **even** when the learning rate is infinitesimal;
2. **Technical**: We derive **novel first**-order SDEs that **remediate** the limitations of classical models;
3. **Insights**: Under a previously unexplored combination of general regularity and noise assumptions, we derive convergence bounds for the SDE models of DCSGD and DSignSGD. These reveal that DSignSGD **remains stable** even under heavy-tailed noise, while DCSGD requires a normalization factor determined **jointly** by the compression rate, noise variance, and smoothness parameters.

## **What the reviewers appreciated:**


### **Presentation and Soundness**
zQGA scored both as **"excellent"**, stating that "[t]he manuscript is **technically strong**, with a **clear structure**, consistent notation, and **effective conceptual explanations**": The Reviewer found that we offer a "**rigorous** and **well-developed** theoretical exploration of distributed stochastic learning". In particular, "[t]he connection between theoretical results and empirical observations [...] is **clearly articulated** and helps bridge formal analysis with intuition". On the same note, rkAo confirms that the "structure is readable", that we state our assumptions **well**, and that our "[p]roofs are straightforward", echoing ahk4 who judged them "**rigorous**".


### **Contributions and Novelty**
We provide a "**detailed** theoretical study" which "highlights the **limitations** of conventional first- and second-order SDE approximations" (zQGA). To remediate this, we "propose *corrected* continuous-time model[s] for DCSGD and DSignSGD" (rkao) "that **correctly** capture learning rate constraints [and] stability thresholds" (zQGA). These provide "**insightful** conclusions on the interplay of different factors" (ahk4) such as "gradient noise, compression, and adaptivity" (zQGA). Ours is a "**unified theoretical** framework" (cisk) that "derives error bounds for a few settings" (Rkao) that "**no prior** [SDE-based] work has worked" with (cisk). To summarize, "the paper presents new [models] that better reflect the results of discrete analysis under a more general $(L_0,L_1)$-smoothness condition and for affine variance/heavy-tailed noise" (9dpY).


### **Experimental and practical perspective**
Several reviewers highlighted potential **"practical guidance"** (zQGA and cisk), especially on a "very practical and efficient method" (9dpY) such as DSignSGD. zQGA requested that we **articulate** this discussion more and complement it with more realistic experiments: We provide this in our revised version. Importantly, we trained an **MLP** (See Figure C.1 and Appendix C.3 for implementation details). These new results exhibit the phenomena prescribed by our theory and already confirmed in our original Figure 1.

## **Main revisions in response to the Reviewers**

We take all criticisms very **seriously** and have carefully revised the manuscript in response. In brief, we have:

1. **Clarified the SDE perspective and positioning**, elaborating more on why we use continuous-time models and why capturing learning-rate restrictions is important; we now also provide a careful, step-by-step justification of the approximation order and validity of our SDEs, clarify their relation to the framework of Li et al. (2017), and added technical details in the proofs.

2. **Specified the scope more explicitly**, including our focus on empirical risk minimization rather than generalization;

3. **Strengthened the practical discussion**, making the implications for DCSGD and DSignSGD more explicit and adding an **additional MLP experiment** (Figure C.1, Appendix C.3) that confirms, in a more realistic setup, the phenomena predicted by our theory and already visible in Figure 1;

4. **Added technical details**, including additional intermediate proof steps, small notation, and readability refinements.

These changes primarily concern **clarification, emphasis, and positioning**, rather than modifications to the core framework or the main conclusions. All changes with respect to the original submission are highlighted in *light brown* in the revised manuscript.

Once again, we are thankful to the reviewers for their constructive feedback. We look forward to the upcoming author–reviewer discussion period.

Thank you for your attention.

Best regards,

The Authors

---

### Meta-Review · Area_Chair_jGbY · 2026-01-04

**Summary:**

This paper proposes a new SDE model for the generalized smoothness setting to address the limitation of the traditional SDE model.

The reviewers acknowledge that the proposed SDE model for handling generalized smoothness is novel. However, they also raise multiple concerns. First, many details are not clearly presented, leading to confusion among several reviewers regarding the proposed SDE model. Although the authors provided additional explanations in the rebuttal, some issues remain unresolved. A substantial revision of the writing is strongly recommended to improve clarity and readability. Second, the proposed SDE model lacks approximation guarantees, whereas existing models provide such guarantees. As a result, it is unclear how well the proposed model theoretically approximates the optimization dynamics. Third, the evaluation is very limited. Although an additional experiment was provided in the rebuttal, it remains small-scale and insufficient to adequately verify the performance of the proposed approach.

These key concerns remain after the rebuttal; therefore, acceptance is not recommended.

**Reviewer Concerns:**

Reviewers raise multiple concerns. First, many details are not clearly presented, leading to confusion among several reviewers regarding the proposed SDE model. Although the authors provided additional explanations in the rebuttal, some issues remain unresolved. A substantial revision of the writing is strongly recommended to improve clarity and readability. Second, the proposed SDE model lacks approximation guarantees, whereas existing models provide such guarantees. As a result, it is unclear how well the proposed model theoretically approximates the optimization dynamics. Third, the evaluation is very limited. Although an additional experiment was provided in the rebuttal, it remains small-scale and insufficient to adequately verify the performance of the proposed approach.

**Reviewer Scores:**

The key concerns remain, so there is no change in the rating.

---

### Decision · Program_Chairs · 2026-01-26

Reject